# Gamma entrainment induced by deep brain stimulation as a biomarker for motor improvement with neuromodulation

Varvara Mathiopoulou [1], Jeroen Habets[1,2], Lucia K. Feldmann [1], Johannes L. Busch [1,2], Jan Roediger[1,2], Jennifer K. Behnke[1], Gerd-Helge Schneider[3], Katharina Faust[3] & Andrea A. Kühn [1,4,5,6] ✉

Finely tuned gamma (FTG) oscillations from the subthalamic nucleus (STN) and cortex in Parkinson's disease (PD) patients undergoing deep brain stimulation (DBS) are often associated with dyskinesia. Recently it was shown that DBS entrains gamma activity at 1:2 of the stimulation frequency; however, the functional role of this signal is not yet fully understood. We recorded local field potentials from the STN in 19 chronically implanted PD patients on dopaminergic medication during DBS, at rest, and during repetitive movements. Here we show that high-frequency DBS induced 1:2 gamma entrainment in 15/19 patients. Spontaneous FTG was present in 8 patients; in five of these patients dyskinesia occurred or were enhanced with entrained gamma activity during stimulation. Further, there was a significant increase in the power of 1:2 entrained gamma activity during movement in comparison to rest, while patients with entrainment had faster movements compared to those without. These findings argue for a functional relevance of the stimulation-induced 1:2 gamma entrainment in PD patients as a prokinetic activity that, however, is not necessarily promoting dyskinesia. DBS-induced entrainment can be a promising neurophysiological biomarker for identifying the optimal amplitude during closed-loop DBS.

Deep brain stimulation (DBS) is an established treatment option for patients with Parkinson's disease, especially those with pronounced motor fluctuations[1]. The latest generation of commercially available IPGs enable the recording of local field potentials (LFPs) from the DBS target regions during active stimulation in chronically implanted patients[2]. This setup allows for the chronic evaluation of electrophysiological biomarkers, such as beta band activity (13–35 Hz) that has been explored in acute settings over the last decades, to develop feedback signals for closed-loop DBS[3].

Within this framework, pathologically enhanced beta band has been related to bradykinesia and rigidity in PD and is considered an antikinetic activity[4,5]. Challenges of current beta band-based feedback algorithms include not only artefact suppression but also modulation of this biomarker with movement, dopaminergic medication, and DBS[6]. Recently, LFPs recorded during high-frequency DBS in patients with PD unveiled an entrained neural signal within the gamma band (40–100 Hz) in both cortical and subcortical recordings. DBS-induced entrainment was identified in a few patients as a narrow-band activity

[1]Department of Neurology, Charité-Universitätsmedizin Berlin, corporate member of Freie Universität Berlin and Humboldt-Universität zu Berlin, Berlin, Germany. [2]Berlin Institute of Health at Charité-Universitätsmedizin Berlin, Berlin, Germany. [3]Department of Neurosurgery, Charité-Universitätsmedizin Berlin, corporate member of Freie Universität Berlin and Humboldt-Universität zu Berlin, Berlin, Germany. [4]Berlin School of Mind and Brain, Charité-Universitätsmedizin Medicine, Berlin, Germany. [5]NeuroCure Clinical Research Centre, Charité-Universitätsmedizin, Berlin, Germany. [6]DZNE, German Center for Degenerative Diseases, Berlin, Germany. ✉e-mail: andrea.kuehn@charite.de

with a peak at ½ of the DBS frequency during the on-dopaminergic state[7]. Based on model predictions, 1:2 entrainment is hypothesized to appear only in certain combinations of DBS amplitude and frequency[8,9] however, its functional role is largely unexplored. Broad band gamma activity (40–100 Hz) in subcortical or cortical LFPs has been closely linked to the initiation and execution of movements and is considered to have a prokinetic effect[10]. In PD patients, a more narrow frequency band (70–90 Hz), referred to as spontaneous finely tuned gamma (FTG), is associated with levodopa-induced dyskinesia (LID), a condition in which PD patients experience involuntary movements as a result of long-term treatment with dopaminergic medication[7,11,12]. Here, we use the term 'spontaneous' to indicate that this activity occurs at rest, in contrast to the movement-related broad band gamma event-related synchronization (ERS). Moreover, it is different from the DBS entrained neural activity. Neural entrainment describes the phenomenon by which ongoing brain oscillations synchronize to the rhythmic pattern of an external stimulus, such as repetitive auditory cues, visual signals, or electrical stimulation[13–17]. Successful neural entrainment by an external stimulus enhanced the cognitive or motor function that was associated to the oscillatory activity being entrained[18,19]. Additionally, DBS can induce evoked responses which can be detected both cortically[20,21] and subcortically[14,22]. Cortical responses have been associated to the hyperdirect pathway while subcortically recorded evoked responses, namely evoked resonant neural activity (ERNA), have been suggested to be elicited by synaptic dynamics within the indirect pathway[23].

Given the role of gamma band activity in a variety of motor phenomena, it is important to understand how DBS modulates ongoing gamma band oscillations, and what is the functional relevance of a potentially entrained signal[24,25]. Here, we make use of the latest DBS pulse generators to study the newly discovered phenomenon of entrained gamma activity in a first cohort study of chronically implanted PD patients under simultaneous dopaminergic treatment and high-frequency DBS. First, we show that STN gamma entrainment occured in the majority of PD patients (15/19) when stimulated with therapeutic relevant DBS intensity. Entrained activity occurred strictly at ½ stimulation frequency, and its amplitude depended on both the stimulation intensity and frequency. Interestingly, spontaneous FTG

activity, when present, gradually entrained to the ½ DBS frequency with increased DBS intensity. For the first time, we were able to correlate entrained activity and kinematic and clinical parameters showing that repetitive finger tapping increased or induced gamma entrainment, and patients with gamma entrainment had a better motor performance. This suggests that entrained gamma activity has a functional prokinetic role. Our results are important both for understanding the neural mechanisms during high-frequency DBS, as well as for clinical practice and future adaptive stimulation.

## Results

### DBS can entrain STN gamma activity in the majority of PD patients depending on stimulation amplitude and dopamine but is not necessarily accompanied by dyskinesia

We found 1:2 gamma entrainment in 15 out of 19 PD patients (10 female, mean age = 62.6 years ±6.7, mean disease duration = 10.9 ± 3.7 years, 19 STNs) treated with STN DBS who were enrolled during standardized 3 or 12-month follow up visits at the Department of Movement Disorders and Neuromodulation. Gamma entrainment in an example LFP recording is shown in Fig. 1. Eight patients displayed spontaneous FTG activity ON medication without stimulation (53%, Patients #1-8, peak frequency μ = 78.4 ± 4.3 Hz). All these patients showed gamma entrainment with stimulation. Additionally, seven patients without spontaneous FTG developed a 1:2 gamma entrainment with increase in stimulation amplitude during ipsilateral DBS. The four remaining patients (21%) showed no 1:2 entrainment (#16–19, for an overview see Fig. 2b). Thus, 79% of PD patients showed STN gamma entrainment at variable stimulation amplitude during DBS. The entrainment occurred at a mean stimulation amplitude of 2.19 ± 0.75 mA and disappeared when DBS was switched off (Fig. 2a). Moreover, the entrainment disappeared in two cases (#2 and #14) with higher stimulation intensities and was reduced in amplitude in one case (#1). Gamma entrainment was restricted to the STN that was stimulated and did not transfer to the non-stimulated other hemisphere in any of our cases.

The patients showing spontaneous FTG developed 1:2 entrainment at a mean DBS amplitude of 1.91 ± 0.68 mA and the patients without spontaneous FTG at μ = 2.6 ± 0.62 mA, which was

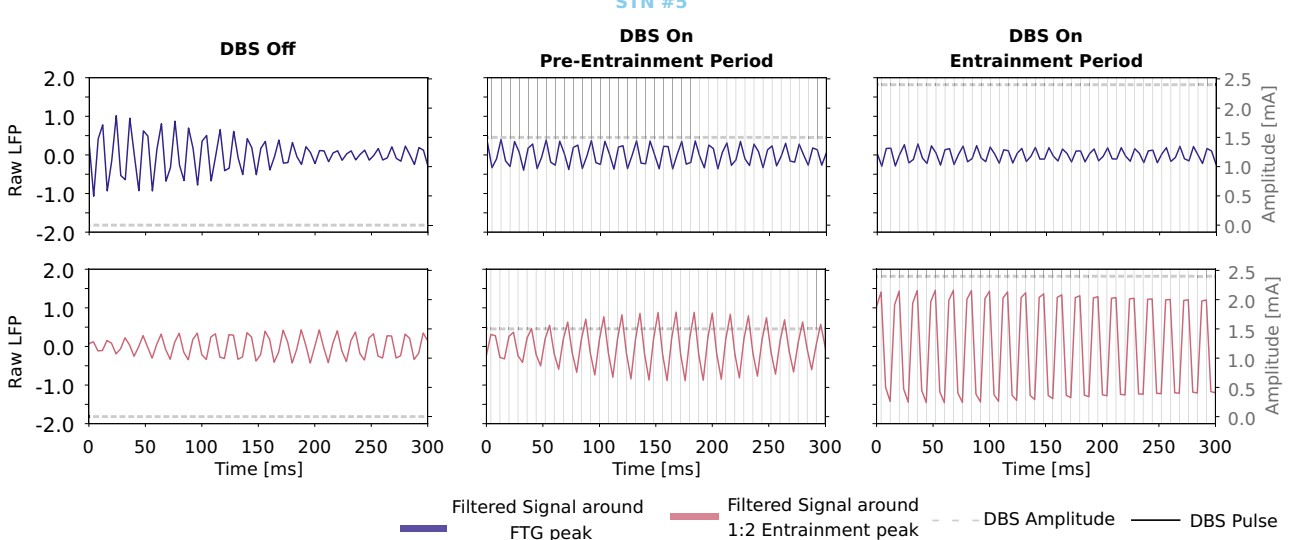

**STN #5**

**DBS Off** | **DBS On Pre-Entrainment Period** | **DBS On Entrainment Period**

Filtered Signal around FTG peak — Filtered Signal around 1:2 Entrainment peak — DBS Amplitude — DBS Pulse

**Fig. 1 | Gamma entrainment in an example LFP recording.** 300 ms filtered 5 Hz around FTG peak (76 Hz in purple) and around the entrainment peak (63 Hz in pink) during DBS-off, pre-entrainment period (DBS amplitude: 1.5 mA), and during highest DBS amplitude where entrainment was observed (2.5 mA). The recovery of the actual DBS pulses was not possible due to hardware constrains. DBS was applied with a frequency of 125 Hz and pulse width of 60 μs, therefore estimated DBS pulses are plotted every 8 ms. During 1:2 entrainment it is visible that a peak of activity appears every two DBS pulses. Waveforms of evoked responses could not be recorded due to the hardware constrains. The respective spectrogram for this patient is presented in Supplementary Fig. 1A, STN #5.

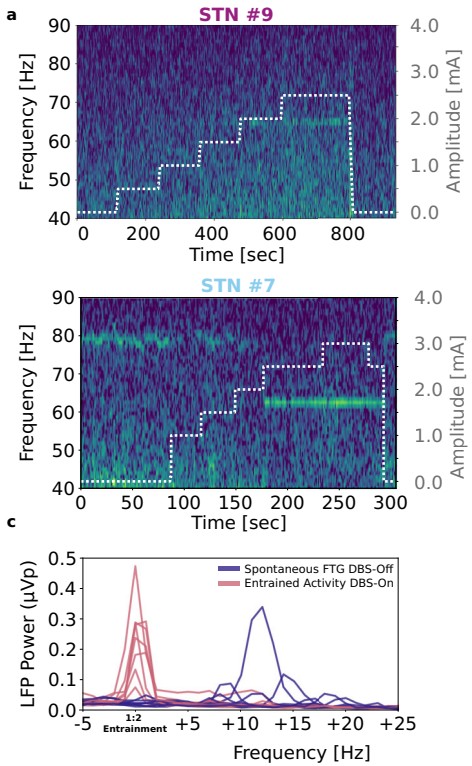

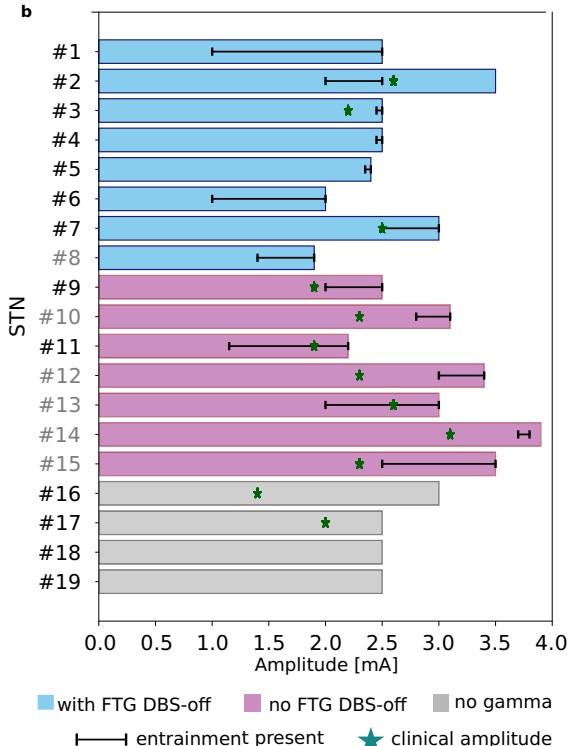

**Fig. 2 | DBS-induced entrainment is modulated by the DBS amplitude.**
**a** Example cases of entrainment at 65 Hz induced by DBS in the absence (upper plot) or in the presence (lower plot) of spontaneous FTG activity during DBS-off. Entrainment appears with increased amplitude and disappears when DBS is switched off. When there is spontaneous FTG during DBS off (lower plot, here around 77 Hz), this adjusts to ½ of DBS frequency, and it resets when DBS is off again at the end of the recording. **b** Schematic presentation of all recordings included in the study. Bar plots along the x-axis show the highest amplitude that was used for every patient. Black lines show the amplitude range in which the entrainment was present. Recordings from STNs #1-8 showed spontaneous FTG in the absence of DBS.

Recordings from STNs #9-15 did not show spontaneous FTG in the absence of DBS. Recordings from STNs #16-19 did not show any activity within the gamma band on/ off DBS. Green stars show the amplitude that was selected during DBS optimization as clinically most beneficial in those cases where the clinically used electrode contact was the same as the one that was used for stimulation during LFP recordings. Grey STN indices (e.g., #8) refer to the STNs that were stimulated directionally. **c** Mean power spectra of 30 sec of rest on/off DBS from all eight STNs in which there was spontaneous FTG during DBS-off. Peaks during entrainment have higher amplitude and are narrower in comparison to peaks during DBS-off.

not significantly different ($P = 0.162$). Regarding clinical features, three out of eight patients with spontaneous FTG exhibited dyskinesia ON medication without stimulation. Two additional patients from this group developed dyskinesia during stimulation, and in two patients dyskinesia increased with stimulation. Patients without spontaneous FTG did not develop dyskinesia during stimulation despite gamma entrainment. UPDRS III improvement with levodopa challenge during recordings was larger in the group with spontaneous FTG (47.8%) as compared to those without (24.7%; $P = 0.024$) pointing to effective dopaminergic treatment promoting entrainment. UPDRS III motor score improved in all patients with DBS (Med On-DBS Off $\mu = 29.72 \pm 13.34$, Med On=DBS On $\mu = 17.47 \pm 10.07$, $P = 0.007$). With respect to entrainment, there was no significant difference in the On-medication motor improvement with DBS using standard clinical amplitude settings between the subgroup of patients that exhibited entrainment and the subgroup without entrainment both in total scores, as well as hemibody subscores. Nevertheless, the test is limited by the small sample size of the group without entrainment ($N = 4$). The DBS amplitudes that resulted in entrainment in each STN are shown in Fig. 2b. Clinical characteristics are shown in Table 1.

## DBS-induced gamma entrainment interacts with spontaneous FTG and depends on stimulation frequency

Properties of non-linear dynamic systems, and specifically the framework of Arnold tongues[8], assume that there is an individual region in

which combinations of stimulation amplitudes and frequencies will have more likelihood to induce 1:2 entrainment (Fig. 3a). We inspected the adjustment of 1:2 entrainment frequency when different DBS frequencies were applied (i.e., 110, 125, and 145 Hz) in five patients, in one hemisphere per patient. Peak frequency of entrained activity followed a strict 1:2 pattern. Specifically, four out of five patients showed 1:2 entrainment that adjusted to 55, 63, and 73 Hz respectively during all three DBS frequencies (Fig. 3a). One patient showed 1:2 entrainment only during DBS at 145 Hz. There was no significant difference in the stimulation amplitude needed for entrainment onset when different frequencies were applied. However, entrained gamma power was smaller with higher stimulation frequency, although the small sample size did not allow for a statistical analysis (110 Hz $\mu = 0.08 \pm 0.06$ µVp; 125 Hz $\mu = 0.03 \pm 0.005$ µVp, 145 Hz $\mu = 0.02, \pm 0.003$ µVp). In line with the idea of an optimal individual region for entrainment, entrained gamma power diminished and disappeared with increased DBS amplitude in one and two patients, respectively.

We further explored whether spontaneous FTG modified the optimal entrainment region in our patients. Eight patients showed spontaneous FTG in the absence of DBS and allowed us to investigate the interaction between spontaneous FTG and DBS-induced entrainment in the gamma band during increasing stimulation amplitudes. Figure 3b shows three averaged envelopes of the analytic signals over time ($n = 8$ STNs): signal around their individual spontaneous FTG peaks, 1:2 entrainment frequency peaks, and the intermediate gamma range between the spontaneous FTG and the 1:2 entrainment

# Table 1 | Demographic and clinical details

| | Patient | Age (y) | Sex | DD (y) | Follow-Up (months) | LEDD (mg) | Electrode | Stim Contact | Recording Contacts | Contacts for chronic stimulation | Gamma Peak DBS-off (Hz) | Entrainment Onset (mA) | UPDRS-III Med Off-Stim Off | UPDRS-III Med On-Stim Off | UPDRS-III Med On-Stim On | Dyskinesia DBS Off/On |
|---|---|---|---|---|---|---|---|---|---|---|---|---|---|---|---|---|
| **WITH FTG DBS-OFF** | #1 | 60–65 | f | 10 | 3 | 775 | 3389 | L2 | L1-3 | 1 | 77 | 1.0 | 22 | 5 | 5 | No/No |
| | #2 | 50–55 | f | 13 | 6 | 300 | 3389 | L1 | L0-2 | 1 | 81 | 2.0 | 78 | 34 | 23 | Mild/Mild |
| | #3 | 50–55 | m | 7 | 3 | 500 | 3389 | R1 | R0-2 | 1 | 73 | 2.5 | 51 | 28 | 22 | No/No |
| | #4 | 60–65 | m | 15 | 3 | 300 | 3389 | R2 | R1-3 | 1 | 73 | 2.5 | 39 | 11 | 11 | Mild/Moderate |
| | #5 | 70–75 | f | 12 | 12 | 500 | SenSight | R2 | R1-3 | 3 | 76 | 2.4 | 44 | 17 | 11 | No/Mild |
| | #6 | 50–55 | m | 13 | 12 | 0 | SenSight | L2 | L1-3 | 1,3 | 84 | 1.0 | 43 | 30 | 7 | No/Moderate |
| | #7 | 50–55 | m | 14 | 3 | 535 | SenSight | L1 | L0-2 | 1 | 80 | 2.5 | 19 | 15 | 9 | No/No |
| | #8 | 65–70 | f | 7 | 18 | 275 | SenSight | R2a | R1-3 | 1a,2a | 83 | 1.4 | 53 | 43 | 12 | Moderate/Severe |
| **NO FTG DBS-OFF** | #9 | 65–70 | m | 11 | 12 | 1250 | 3389 | L2 | L1-3 | 2 | (-) | 2.0 | 54 | 44 | 33 | No/No |
| | #10 | 70–75 | m | 20 | 12 | 1000 | SenSight | R2b | R1-3 | 2 | (-) | 2.8 | 49 | 39 | 31 | No/No |
| | #11 | 70–75 | f | 7 | 12 | 475 | SenSight | R1 | R0-2 | 1a,1c | (-) | 1.5 | 29 | 19 | 12 | No/No |
| | #12 | 70–75 | f | 14 | 12 | 375 | SenSight | R1b | R0-2 | 0,1 | (-) | 3.0 | 51 | 39 | 21 | No/No |
| | #13 | 55–60 | m | 15 | 12 | 550 | SenSight | R1c | R0-2 | 1 | (-) | 2.0 | 34 | 27 | 23 | No/No |
| | #14 | 55–60 | m | 7 | 12 | 350 | SenSight | R2a | R1-3 | 2a,2b,3 | (-) | 3.7 | 57 | 35 | 11 | No/No |
| | #15 | 65–70 | f | 7 | 12 | 400 | SenSight | L2b | L1-3 | 2b,2c | (-) | 2.5 | 42 | 35 | 27 | No/No |
| **NO GAMMA** | #16 | 60–65 | f | 7 | 3 | 525 | 3389 | L2 | L1-3 | 2 | (-) | (-) | 42 | 26 | 14 | No/No |
| | #17 | 60–65 | f | 7 | 12 | 0 | 3389 | R2 | R1-3 | 2,3 | (-) | (-) | 40 | 28 | 10 | No/No |
| | #18 | 55–60 | m | 11 | 12 | 200 | 3389 | R1 | R0-2 | 2 | (-) | (-) | 38 | - | 9 | No/No |
| | #19 | 65–70 | f | 11 | 12 | 1000 | 3389 | R1 | R0-2 | 0,1 | (-) | (-) | 67 | 60 | 41 | No/No |

*DD* Disease duration, *LEDD* Levodopa equivalent daily-dose.

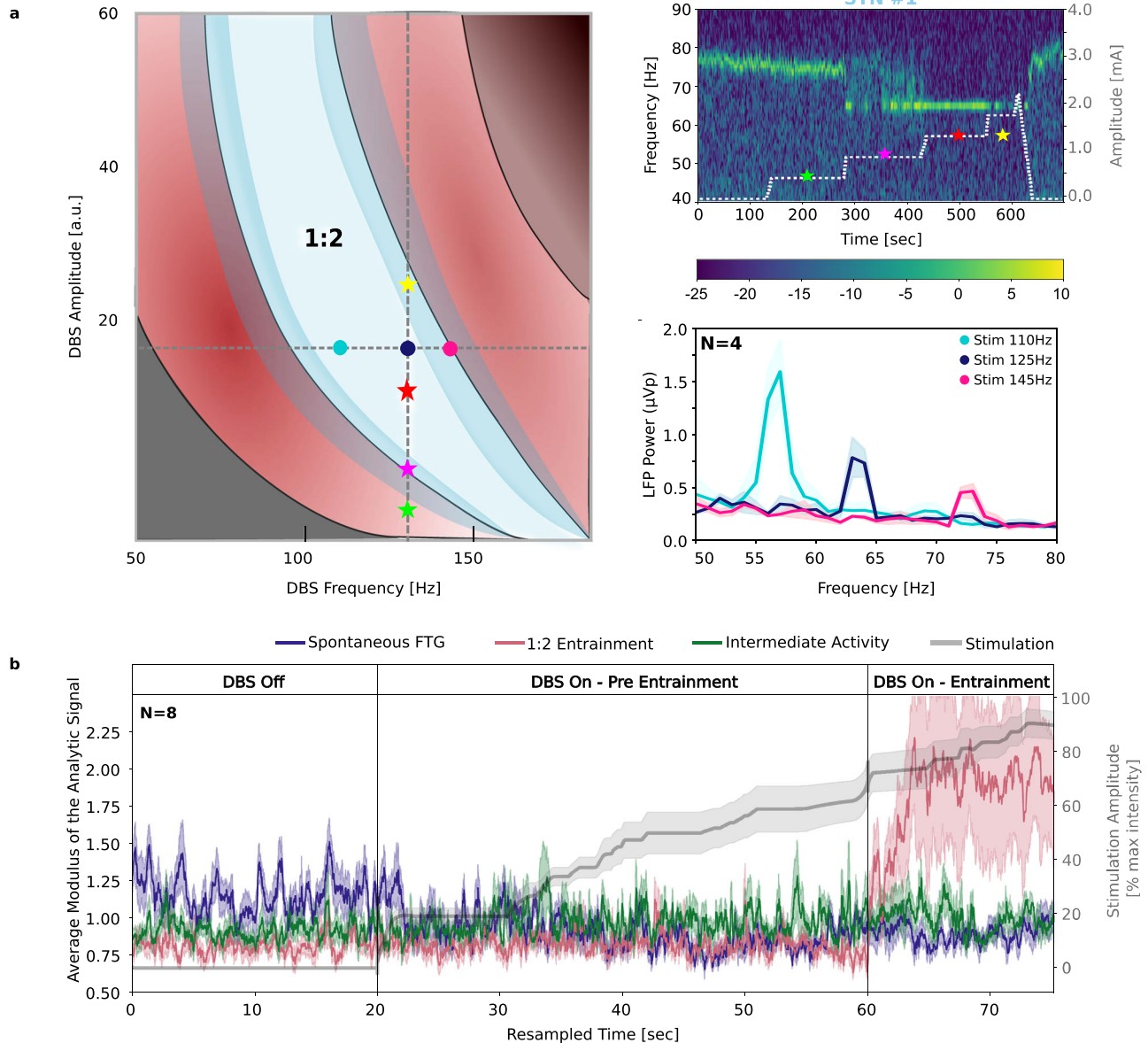

**Fig. 3 | DBS-induced entrainment interacts with spontaneous FTG. a** Left: Schematic representation of Arnold tongues, explaining the occurrence of 1:2 entrainment depending on deep brain stimulation (DBS) frequency and amplitude. At a stable DBS frequency (dashed vertical line indicates 130 Hz), 1:2 entrainment results from a certain range of DBS amplitudes (marked by the white area and the red star). Amplitudes in the lower or upper borders of this area (yellow and magenta stars, respectively) have a lower likelihood of causing 1:2 entrainment. The varying borders of the white area represent the dynamical nature of these borders based on neural noise. Amplitudes outside of the white area (e.g., the green star) do not cause 1:2 entrainment. This concept is depicted in the right spectrogram: Spectrogram of exemplary patient stimulated continuously at 130 Hz. The stars on the spectrogram correspond to different DBS amplitudes falling within distinct regions of the frequency-amplitude space defined by the Arnold tongue. At 1.0 mA,

1:2 entrainment is not fully present, but at 1.5 mA, it becomes noticeable, disappearing again at 2.0 mA. Similarly, at a stable DBS amplitude (dashed horizontal line in left illustration), there is a varying likelihood of 1:2 entrainment to be induced by different DBS frequencies. DBS frequency at 110, 125, and 145 Hz are denoted with turquoise, blue, and magenta dots respectively. This is shown in group power spectra of 1:2 entrainment in different DBS frequencies (right bottom plot). Shaded areas around the power spectra denote the standard error of the mean. **b** Averaged envelope of the analytic signal (STNs #1-8) of spontaneous FTG peak as identified in the DBS-off state (blue), of the 1:2 entrainment frequency band as identified during DBS-ON (pink), and of the intermediate activity (green) with increased DBS intensity (grey). All recordings were resampled in equal epochs denoted by the solid vertical lines. Shaded areas show the standard error of the mean.

frequencies. There is a gradual decrease of spontaneous FTG frequency shortly after switching DBS on ($P < 0.001$). With higher stimulation amplitude we observed a period when gamma activity was fluctuating within the intermediate frequency range (i.e. between spontaneous FTG and entrained activity) that continues until stable entrainment frequency is reached with further increase in stimulation amplitude. This transition period is revealed by an increase of intermediate activity before entrainment in comparison to the DBS-off period ($P < 0.001$).

In the same group we compared peak amplitudes and widths of spontaneous FTG and 1:2 entrainment during rest. Peaks during entrainment had higher amplitudes ($P = 0.049$) and narrower widths ($P < 0.001$) than peaks during spontaneous FTG (see Fig. 2c). There was no correlation between the power of entrainment and the difference between the spontaneous FTG peak and the entrainment peak frequencies ($P = 0.365$) suggesting that power of entrainment does not relate to FTG frequency although stimulation could not be performed at specific FTG frequency due to hardware limitations.

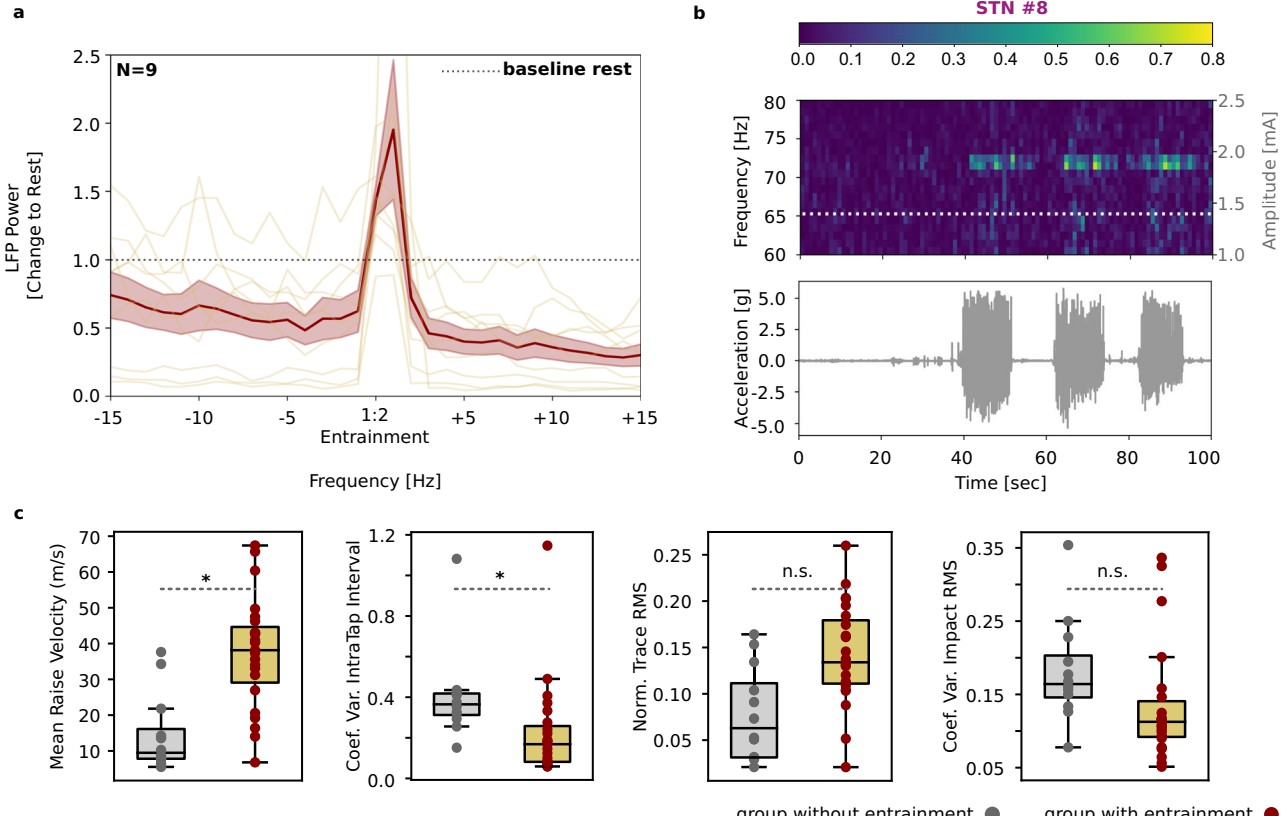

**Fig. 4 | DBS-induced entrainment is modulated by movement and is present in patients that had better performance in a motor task. a** Mean power spectrum of 9 STNs during movement corrected to rest period. Values > 1 indicate higher amplitude of entrainment during movement. Shaded area around the spectrum shows the standard error of the mean. **b** Upper plot: Spectrogram of an exemplary case of STN stimulated at 1.4 mA and 145 Hz. Bottom plot: synchronized movement traces of three blocks of finger tapping lasting 10 s each, with 10 s rest in between. Entrainment persists during the blocks of repetitive finger tapping but fades away directly after the movement is stopped. **c** Comparisons of the different movement metrics between the group with entrainment (*n* = 9 STNs) and the group without (*n* = 4 STNs). LME was used to test for differences between the groups, while taking

into account for the repeated measures. It is shown that the group that exhibited entrainment performs better in repetitive finger tapping than the group that did not (mean raise velocity *P* = 0.017, coefficient of variation of intra-tap interval *P* = 0.043, normalized RMS *P* = 0.058, and coefficient of variation of the impact force *P* = 0.058). DBS amplitude had no significant effect as a fixed effect on these four metrics in Linear Mixed Effects Models. The box length is the interquartile range and the whiskers extend to the minima/maxima of 1.5 x the interquartile range respectively. The lower/upper bound of the box is the 25th/75th percentile respectively. Median is denoted with a horizontal line within the boxplot. Outliers are plotted individually.

## DBS-induced gamma entrainment is modulated by movement and is linked to successful motor performance

We further investigated the functional relevance of DBS-induced entrainment by testing whether it is modulated by movement or influences motor performance. Nine out of 15 patients performed a finger tapping task during the DBS settings that evoked entrainment. In these patients, averaged power of the 1:2 entrainment increased during the finger tapping movements compared to rest (Fig. 4a; Power during movement, baseline-corrected to rest μ = 1.41 ± 0.33 μVp, t(8) = −20.67, *P* = 0.008). Moreover, in an exemplary case we observed the presence of 1:2 entrainment only during movement (see Fig. 4b). In the OFF-DBS condition, a similar increase in spontaneous FTG power during movement was observed in two patients that exhibited FTG (data not shown).

We also compared the motor performance during DBS between the group that showed entrainment (*n* = 9) and the subgroup that did not (*n* = 4). Mean raise velocity was higher (*P* = 0.017, Confidence Intervals CI: 3.705-37.59) and the coefficient of variation of intra-tap interval was lower in the entrainment group (*P* = 0.043, CI: −0.358:-0.006), indicating better tapping performance. Normalized RMS of the trace shows a trend (*P* = 0.058, CI: −0.007-0.123) and coefficient of variation of the impact force did not yield into statistical significance (*P* = 0.058, CI: −0.007-0.123, P = 0.184, CI: −0.126-0.024). DBS

amplitude had no significant effect as a fixed effect on these four metrics in Linear Mixed Effects Models (LMEs; Fig. 4c). There was no significant correlation between the power of entrainment during movement and the improvement in the motor task on the above metrics pointing to a step-wise increase in entrained gamma activity with movement.

## Discussion

Here, we investigate the DBS-induced 1:2 gamma entrainment in subthalamic recordings in a large cohort of 19 patients with Parkinson's disease. We showed that about 80% of PD patients ON-medication present DBS-induced gamma entrainment during therapeutic high-frequency DBS. Particularly, we describe for the first time systematically the electrophysiological characteristics of entrained gamma activity in relation to clinical observations and show that gamma entrainment is related to improved motor performance but not necessarily to dyskinesia. We confirmed that entrainment appears with increased DBS amplitude at exactly half the frequency of the DBS frequency applied, which was irrespective of the presence of spontaneous FTG without stimulation or the occurrence of dyskinesia. If spontaneous FTG was present, we show that there was a transition period by which, with increasing DBS amplitude, the peak frequency of spontaneous FTG gradually decreased until it locked to ½ DBS

frequency, which further speaks for an entrained gamma being a non-artefactual phenomenon. In the same vein, in a subset of patients the power of 1:2 entrainment was reduced with higher DBS amplitude or frequency. Further, our results show an enhancement of the 1:2 entrainment signal caused by short finger-tapping blocks, and patients with gamma entrainment had better motor performance as compared to those without entrainment, pointing at a prokinetic role within the motor network. These findings suggest that there are previously unexplored mechanisms of DBS and our data confirm previous models of gamma entrainment with DBS[21] that we will further discuss in the context of movement and DBS therapy in PD.

## DBS-induced gamma entrainment is not merely an artefact

DBS-induced 1:2 gamma entrainment has been described only recently in recordings from the STN and cortex in PD patients[7,26,27] and model-based approaches have confirmed subharmonic entrainment of neural oscillations[8,9,28]. Such subharmonic activity should be treated cautiously as a potential artefact. Several features, such as medication dependency, modulation with sleep stage, and presence of dyskinesia, as well as non-linear stimulation response have been discussed supporting its physiological origin, however, its functional role is still not well understood[28]. Here, we confirm the model predictions in a large patient group and provide further evidence that argue against the entrainment merely being an artefact of DBS. First, we showed that DBS-induced gamma entrainment interacts with the spontaneous FTG by gradually locking the FTG to half of the DBS frequency with increasing amplitudes (Fig. 3). This observation fits the hypothesis that entrainment is facilitated by the presence of endogenous oscillatory activity belonging to physiological processes[13,27,29,30]. Second, increased DBS amplitudes led to an attenuation of the entrainment in three out of 15 cases (Fig. 2b), a finding that confirms a non-linear response to stimulation. This attenuation of 1:2 entrainment with higher DBS intensity fits the Arnold tongue framework which suggests that entrainment only appears in certain regions of the amplitude-frequency space. In line with this, the amplitude of entrained activity was also modulated by frequency of stimulation (Fig. 3a)[8]. Third, entrainment was modulated by movement and related to better motor performance supporting a prokinetic functional role of entrained gamma activity. Evoked potentials and resonant neuronal[23] activity that have been described in the STN-GPi loop have to be considered as also causing entrained gamma activity. However, constrains of the hardware filters when using Percept limit any further analysis of high frequency and evoked activity.

## DBS-induced entrainment is modulated by movement and is present in patients that had better performance in a motor task

So far, FTG has been described as an oscillatory activity with a peak frequency between 60 and 90 Hz in the motor network in different basal ganglia nuclei[31], the thalamus[32], and the motor cortex[7], and has been associated with level of alertness and increased motor activity. This was distinguished from a broadband gamma activation that occurs with movement and most likely reflects neuronal spiking activity over a broad frequency range (30–200 Hz)[33]. We observed an increased amplitude of gamma entrainment during movement that supports its functional relevance. Studies in different neurophysiological fields showed that entrainment requires both an endogenous oscillatory activity as well as an external stimulus. Specifically, it was shown that repetitive high-frequency stimulation enhanced local ongoing activity and that entrainment was more prominent when the stimulus was applied in frequencies closer to the intrinsic frequencies[13,18–21,27,29]. This external manipulation of ongoing oscillatory dynamics was also found to have beneficial effects in processes such as associative learning, and to improve medical conditions, such as tinnitus[14,18].

Entrainment caused by subthalamic DBS would typically appear in the gamma band at stimulation frequency (i.e., 130 Hz) or its sub-harmonics (65 Hz). Activity in the gamma band increases with voluntary movements (40–90 Hz) and with effective dopaminergic medication[10] indicating its prokinetic effect. Within this framework, an increase in prokinetic neuronal activity related to dopamine or movement could also facilitate a successful entrainment during stimulation, as observed in our patients.

More importantly, we show that patients that exhibited DBS-induced gamma entrainment performed better in the finger tapping task than the patients that did not. Although entrainment amplitude did not correlate with motor improvement, we hypothesize that entrainment per se could facilitate the physiological mechanism that is related to the 1:2 entrainment frequency, in this case likely the prokinetic gamma synchronization. This hypothesis is supported by previous studies that showed that stimulation at 20 Hz (i.e. within the beta band of 13–35 Hz) worsened motor performance and bradykinesia[34,35]. In this case, entrainment possibly promoted activity within the beta frequency range that is considered pathological and enhanced parkinsonian symptomatology. In our study, high frequency DBS-induced entrainment may enhance physiological oscillatory activity within the broadband gamma, and this may explain part of the prokinetic effect of subthalamic high-frequency DBS and its effectiveness in alleviating bradykinesia. Interestingly, stimulation at individual gamma frequency (70–90 Hz) that could be assumed to lead to 1:1 entrainment has not been described to specifically improve bradykinesia but showed similar results to standard high frequency stimulation[36]. Clinically, 70–90 Hz stimulation is often used to improve dyskinesia, dysarthria or freezing in some patients[37–39]. Further research is needed to clarify the underlying mechanism for 1:2 gamma entrainment, potentially representing an additional mechanism of DBS. Our data suggest a potential utility of 1:2 entrainment as an informative signal when adjusting DBS amplitudes for patients with PD.

## Finely tuned gamma activity in the motor network and dyskinesia

FTG has been observed in PD under dopaminergic medication, and in most cases it was associated with dyskinesia in both human patients and animal models[7,11,25,26,40]. In the framework of pro- and antikinetic activity[41] gamma activity is considered a prokinetic brain rhythm that is observed in the ON-medication condition in parallel with improved motor performance[4]. More recently, DBS entrained gamma band activity has been described mainly in the motor cortex in selected PD cases and used as a biomarker for dyskinesia during closed-loop DBS. In this context, entrained gamma is proposed as a feedback biomarker for adaptive DBS[42]. Interestingly, dyskinesia only occurred in those patients (three out of eight) that presented spontaneous FTG under dopaminergic medication before stimulation was switched on. In four out of eight patients from this subgroup, dyskinesia primarily occurred or worsened with DBS. Patients without spontaneous FTG did not show dyskinesia even though they had gamma entrainment. Nevertheless, these patients showed better motor performance as compared to those without entrained gamma pointing to its functional role within the motor network. Another interesting observation is that patients with spontaneous FTG had a better motor improvement with dopaminergic medication at the time of recording as measured by UPDRS III.

In this context one could hypothesize that dopaminergic medication induced a broader motor network facilitation in this cohort that is represented by spontaneous FTG leading the network towards a state that is prone to dyskinesia. Entrained gamma activity may lead to a similar but more constrained motor network activation that is less likely to overshoot resulting in dyskinesia (intermediate level) but lead to improved motor performance. Only if the network is already at an elevated level of activity, dyskinesia may break through (high level). In

line with this, DBS induced gamma entrainment is less likely to occur in the OFF-dopaminergic medication condition. Depending on the pro-kinetic activation level of the motor system, which is modulated by dopamine, focal network stimulation (STN DBS) can lead to gamma entrainment with improved motor performance. This could poten-tially result in dyskinesia (high level of activation represented by spontaneous FTG) or not (intermediate level) when the gamma activity increase is more rigidly confined within the borders of the sub-harmonic DBS frequency. One could hypothesize that DBS partially exerts its effects through motor network activation confined to a narrow-band gamma frequency. This activity may propagate less across the network but is balanced by the activation of the underlying dopaminergic network state. Reduction in dopaminergic medication with STN DBS is a prerequisite in clinical routine to allow effective stimulation without adverse motor and non-motor effects in PD patients. The specific interplay between broad network state changes induced by dopaminergic medication, and more focal and narrow-band stimulation induced activation pattern need to be characterized in future studies.

### DBS-induced gamma entrainment and closed-loop DBS

DBS-induced gamma entrainment has been recently introduced as a feedback signal for closed-loop DBS[42]. Here, the onset of entrainment has successfully been linked to a high dopaminergic state that allows for reduction of the stimulation amplitude. Entrained gamma is a potential robust closed-loop DBS input signal as it has an entirely predictable frequency and large peak amplitude, which can be also used in chronic devices with limited spectral recording bandwidth. Our data suggest that this biomarker is not solely associated with dyski-nesia but could be assigned to a prokinetic state that should be maintained. The positive trend between gamma entrainment and motor performance we observed in our cohort would support to maintain DBS during gamma entrainment if dyskinesia is absent. A more complex control mechanism taking into account the balance between pro- and antikinetic neuronal biomarkers may help to balance stimulation amplitude within the optimal borders to maintain effective stimulation without dyskinesia. Future studies could focus on recording 1:2 gamma entrainment chronically, while patients are at home and experience all different levels of dopaminergic medication intake, together with other daily activities and dyskinesia severity. This approach could help optimize its potential for use amplitude mod-ulation in closed-loop DBS systems. Additionally, the influence of contact selection for stimulation could be investigated to disentangle other DBS parameters that influence 1:2 entrainment.

### Limitations

We also have to consider some limitations of the current study. First, the patients were recorded under two protocols, which had an impact on the selection of the recording contacts. In patients with 3389 electrodes the contact pairs with the highest beta band activity were used for recording, while in patients with SenSight electrodes contact selection for stimulation was based on gamma entrainment during a previous systematic survey. This precluded further analysis of contact specificity for gamma entrainment but unlikely had a significant impact on the reported results. Second, the stimulation protocol was focused on short-term modulation of neuronal signals by stimulation and its clinical consequences for motor performance. This precluded the assessment of long-term stimulation effects especially on dyski-nesia that sometimes occur only after minutes or hours of high amplitude stimulation. Third, the patient group that did not show gamma entrainment was limited to four patients, which led to an imbalance in power for the comparison between groups. However, the results were consistent across the cohort despite the small sample size, and are supported by previous reports, where enhanced gamma activity is linked to improved motor performance. Fourth, in many

patients the stimulation amplitude was not increased substantially, as it was cut-off to the level in which side-effects were tolerated. This did not allow us to investigate, whether the power of entrainment would decrease with higher stimulation intensities in a larger number of patients (apart from the three cases described here), in line with the assumptions of the Arnold Tongue framework. Finally, levels of dopamine were not modulated within our patient group and stimula-tion frequency could also not be adjusted exactly to the frequency of FTG when present due to hardware limitations of the pulse generator precluding more specific exploration of these network changes on entrainment. Nevertheless, there was no correlation between entrained gamma power and frequency of FTG.

Here, we propose a functional role of DBS-induced gamma entrainment in the STN of patients with Parkinson's disease for motor improvement. We show that the amplitude of the entrainment increased during movement and that motor performance was improved in the group with than in the group without entrainment. Importantly, we propose that entrained gamma activity is more likely a biomarker for a good ON state related to successful motor perfor-mance in PD as compared to the finely tuned gamma activity that is more closely related to dyskinesia. This distinction will be crucial for future application for adaptive DBS. We also show that entrainment interacts with ongoing oscillatory activity, DBS intensity and frequency of stimulation, an observation that is well interpreted by the concept of Arnold tongues. These findings support a prokinetic role of DBS-induced gamma entrainment for PD that may be a mechanism leading to motor improvement with DBS.

## Methods

### Subjects and study protocol

All patients gave written informed consent prior to the study. The study protocol was approved by the local ethics committee (Medical Ethical Committee of Charité Universitätsmedizin, Berlin, Germany, Protocol ID.: EA2/256/20) in accordance with the standards set by the Declaration of Helsinki. Sample size was determined based on experience with previous LFP studies using similar analysis methods. No data were excluded from the analyses. The experiments were not randomized and the investigators were not blinded to allocation dur-ing experiments and outcome assessment.

In all patients the placement of the electrodes was determined by stereotactic planning, intraoperative microelectrode recordings, and clinical testing, and was confirmed by post-operative imaging[43]. The patients were implanted bilaterally with Medtronic 3389 (n = 9) or SenSight (n = 10) DBS electrodes, which were connected to the sensing enabled 'Percept' IPG (Medtronic ©, Minneapolis, MN, USA). Demo-graphic and clinical details are presented in Table 1. We followed the reporting guidelines based on the REMARK checklist[44].

All recordings were performed ON-medication 30 min after patients received 100–200 mg fast-acting L-Dopa. LFPs were recorded during a systematic unilateral ramping of stimulation amplitude at rest and during three runs of 10 seconds repetitive finger-tapping. DBS pulse width and frequency were set at 60 μs and 125 Hz (SenSight electrodes) or 130 Hz (3389 electrodes). Patients performed all recordings seated comfortably in an armchair. All sessions were video recorded. LFPs were recorded bilaterally in a bipolar montage between the two contact-rings adjacent to the stimulation level, according to standard Medtronic Percept settings[45]. Nine patients (all with 3389 circular electrodes) were reported previously[2,6]. In these cases, contact selection was based on the strongest beta activity (13–35 Hz) during OFF-medication and other ring contacts were not tested. Contact selection in the remaining 10 patients (SenSight electrodes) was based on the occurrence of gamma entrainment during a monopolar review in OFF-medication, which is described elsewhere[46]. Stimulation was applied at single segments (n = 6) or at the ring level (n = 4). Addi-tionally, in five of those patients (patients #8, 10, 12, 13, 15) the protocol

was repeated at three different DBS frequencies (i.e., 110, 125, and 145 Hz).

## Data acquisition and analysis

**LFP recordings and analysis.** Continuous time series of LFPs originating from the STN were recorded using the BrainSense Streaming feature of the Percept IPG with a sampling frequency of 250 Hz. LFPs were preprocessed using the open source 'PERCEIVE Toolbox' (utilizing the SPM12 and Fieldtrip packages) and further analysis was performed using Python[47,48]. The stimulation pulses were internally filtered by the hardware, therefore they could only be estimated based on the DBS frequency and pulsewidth. Data were transformed to the time-frequency domain using a Fast Fourier Transform with a window size of one second and a 25% overlap. Individual spectrograms of all patients are presented in Supplementary Fig. 1. Rest activity was isolated during all stimulation conditions for each patient. Gamma peak frequencies were defined as the frequency with the largest amplitude between 40 and 100 Hz. Peaks of gamma in FTG/entrainment were selected visually and were the only distinct peaks with the highest amplitude in 40–90 Hz. The peak selection was confirmed by using 'findpeaks' in Python (threshold = 0.03).

We further investigated the effect of stimulation on three signals filtered around different peaks, i.e., around the patient specific peak of FTG (referred to as 'spontaneous FTG'), around the 1:2 entrainment peak, and taking the intermediate activity between these two. To obtain these three signals, raw data were filtered and averaged 5 Hz around the subject-specific spontaneous FTG peak (e.g., 78 Hz: 76–80 Hz), 5 Hz around the subharmonic frequency at ½ the DBS frequency (e.g., 65 Hz: 63–67 Hz), and between the spontaneous-FTG and entrainment (i.e., in this example 67–76 Hz). The analytic signal was calculated with the Hilbert transform. Next, we took the envelope analytic signal (absolute values), z-scored it to allow for an inter-subject comparison, and smoothed it with a moving average of 2 s. For the analysis, we divided the recordings of subjects with entrainment into three phases: DBS-OFF, DBS-ON before the entrainment onset, and DBS-ON after the entrainment onset. Since these phases have varying durations across patients, we resized each phase to a common duration by resampling the smoothed z-scored envelope of the analytic signal. Resampling the z-scored envelope preserved the overall dynamics of the signal, while allowing for group level analyses. To compare the fluctuation of these three signals, we extracted 2500 samples (10 s) during DBS-Off, 10 s after DBS was switched on, and the last 10 s before entrainment onset.

## Motor assessment

Clinical improvement with dopaminergic medication was assessed as change in Unified Parkinson's Disease Rating Scale Part III (UPDRS III) scores before and after dopaminergic medication intake. The presence and absence of dyskinesia was assessed by a clinician present at the recordings and assessed using the Clinical Dyskinesia Rating Scale[49]. Change in motor performance during stimulation was assessed by finger tapping. To quantify the motor task, a tri-axial accelerometer was attached to the index finger opposite to the side where the unilateral DBS was applied. Accelerometer traces were captured at a sampling frequency of 4 kHz (TMSi Saga) and were down sampled to 250 Hz. The traces were synchronized to the LFPs using a custom-made synchronization device. This device produced a visually detectable signal in the video recordings, while simultaneously generating a detectable vibration in the accelerometer traces.

Movement blocks were considered during highest DBS amplitude with entrainment in the cohort in which entrainment was induced, and during the highest DBS amplitude applied in the cohort without entrainment. Movement traces were further processed and analyzed with the open-source algorithm 'ReTap'[50]. ReTap was used to automatically detect both single taps and ten-second blocks of finger tapping movement. Based on the performance and diversity of kinematic concepts captured, we included the following four movement metrics in our kinematic analysis: normalized root mean square, mean finger raise velocity (m/s), coefficient of variation of intra-tap intervals, and coefficient of variation of root mean square impact. For further details we refer to the ReTap validation study.

## Statistical analysis

A two-sided wilcoxon signed-rank test was used for group comparisons and Spearman rho for correlations. This is a non-parametric test that makes no assumption about the underlying data distribution. A two-sided t-test was used to compare the power of entrainment during movement in comparison to baseline. A linear mixed effect model was used to investigate the relationship between the presence of entrainment and the motor performance, while correcting for the repetition of movement blocks within patients with a random intercept.

## Reporting summary

Further information on research design is available in the Nature Portfolio Reporting Summary linked to this article.

## Data availability

The data that support the findings of this study are available on request from the corresponding author. The data are not publicly available due to GDPR-compliance rules.

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

## Acknowledgements

We would like to thank the patients that participated in the study. We would like to thank Ulrike Uhlig and Rose Franx for their help with organizing and assisting during the recordings. We thank Leon Steiner for fruitful discussion on evoked activity. A.A.K. received funding from the Lundbeck Foundation as part of the collaborative project grant "Adaptive and precise targeting of cortex-basal ganglia circuits in Parkinson´s Disease" (Grant Nr. R336-2020-1035), from the Deutsche

Forschungsgemeinschaft (D.F.G., German Research Foundation) – Project-ID 424778381 – TRR 295, and from the Deutsche Forschungsgemeinschaft (DFG, German Research Foundation) under Germany´s Excellence Strategy – EXC-2049 – 390688087. J.H., J.R., and J.L.B. are fellows of the BIH Charité Junior Clinician Scientist Program.

## Author contributions

V.M., J.H., and A.A.K. conceptualized and designed the study. V.M. conducted the electrophysiological, movement, and statistical analysis and drafted the study. V.M., J.H., L.K.F., performed the electrophysiological recordings. J.H., L.K.F., J.L.B., J.R., J.K.B., and A.A.K., contributed to the interpretation of the data, revised, and edited the work critically. J.L.B., J.R., J.K.B., K.F., and G.H.S. contributed to the data acquisition. A.A.K. contributed critically to the supervision, project organization and funding acquisition. All coauthors have reviewed and approved the content of the manuscript.

## Funding

## Competing interests

A.A.K. received honoraria for consultancies and/or talks from Medtronic, Boston Scientific, and Stada Pharm. L.K.F. received honoraria for consultancies/talks from Medtronic. The remaining authors report no competing interests
