## [Transparent Peer Review file · Nature Communications]

Gamma entrainment induced by deep brain stimulation as a biomarker for motor improvement with neuromodulation

Corresponding Author: Professor Andrea Kühn

Version 0:

Reviewer comments:

Reviewer #1

(Remarks to the Author)

The authors investigate the role of DBS-induced gamma entrainment as a potential biomarker for motor improvement in Parkinson's disease (PD). They report that DBS at therapeutic levels entrains gamma oscillations in the STN at half the stimulation frequency in a majority of patients, which was associated with enhanced motor performance but not necessarily dyskinesia. The results suggest that DBS-induced gamma entrainment as a candidate biomarker for adaptive DBS, with implications for optimizing motor outcomes in PD patients.

- To me the most interesting aspect of this paper is the modulation/interruption or creation of gamma by DBS. What however does 'finely tuned' gamma mean? It seems to place a kind of value judgement on an oscillation that is incompletely defined. What makes it 'fine'? What makes it 'tuned' (I supposed the association with the DBS frequency, but in what way is it tuned when it occurs spontaneously?). Prior work suggesting this is a biomarker for unwanted dyskinesias would not in my mind make it 'fine,' maybe this refers to it being narrow- rather than broad-band?

- The authors provide frequency domain data for neural entrainment by DBS but neglect decades of prior research on DBS-evoked potentials recorded from various sites. Event related potentials measure neural synchronization, in this case to the DBS pulse. The prior work indicates that at some level DBS entrains scalp potentials over ipsilateral motor cortex (Baker KB et al 2002), including at therapeutic high frequencies (Walker HC et al 2012). Recent work also provides evidence for entrainment within both the STN (Sinclair N et al 2018) and the GPi (Awad et al 2021, Johnson K et al 2024).

- The one-half frequency analyses do not evaluate for entrainment at the stimulation frequency, likely because of the large electrical stimulus artifact (~130 Hz is truncated from the y axis). So while interesting in its own right, my suspicion is that this half frequency phenomenon is a network behavior that can become superimposed over entrainment at the stimulation frequency. Single pulses elicit complex waveforms are longer than the interstimulus interval during therapeutic stimulation. As such, the prior stimulation history would likely induce summation effects that could amplify or suppress gamma in complex ways that depend on patient-specific network characteristics, DBS electrode location, stimulation parameters, and other factors. I think the current work would benefit from showing evoked potential waveforms to the stimulus pulse in these patients to get a better sense of how stimulation is driving the network in the time domain.

- One notable strength is that this work is performed with a novel device with recording and stimulation capabilities in chronically implanted patients.

- In any case, DBS-induced gamma entrainment could serve as a novel biomarker for motor improvement, which could provide a parameter to guide closed-loop DBS strategies. This work demonstrates potential new avenues for improving motor performance in PD patients.

- Experimental design and setup is clear with a focus on the relationship between gamma entrainment and motor performance.

- A substantial number of patients did not display the gamma entrainment phenomenon (n=4); presumably these patients improved with DBS. What were their motor improvements? What is the improvement in contralateral UPDRS part 3 sub-scores in patients who did versus did not exhibit entrainment? Is the one half gamma entrainment necessary for

improvement.

- They should correlate the amount of entrainment with the amount of improvement on their motor task.
- These are on medication studies, so a component of the motor performance likely relates to how dopaminergic medications impact individual participants, not related to stimulation alone. The gold standard for understanding motor improvements related to DBS is off dopaminergic medications >12 hours.
- The study focuses on short-term symptomatic effects, limiting insights into the long-term implications of gamma entrainment on motor function and dyskinesia.
- The study does not systematically vary dopamine levels, which could further clarify the interaction between dopaminergic medication and gamma entrainment and how this might relate to a closed loop DBS paradigm.
- In multiple places, the authors say something is changed versus something else, but that it did not reach statistical significance. My view is we have to go by the statistics. It is either significant or not significant at their level of statistical power. As written, they read as interpretations. I don't think the inferences are justified without statistical significance, although they might offer some speculations for future hypothesis testing in the discussion.

Reviewer #2

(Remarks to the Author)

The authors present data on a topic that is of increasing interest to researchers in the field of DBS. Although adaptive DBS approaches are hoped to provide a way to improve patient outcomes with DBS, a better understanding of potential biomarker signals to use in control algorithms is essential. This study adds to our understanding of one such biomarker signal: finely tuned gamma oscillations. The manuscript is generally well written. For the most part data is presented that supports the authors claims, but there are several instances where I believe the authors overstate their findings as indicated in comments below. The claim that FTG entrainment improves motor performance could be reworded or be better supported by a correlation analysis directly relating the amount of entrainment to the degree of motor improvement.

Specific comments according to manuscript section are below.

Abstract

"DBS-induced entrainment can be a promising real-life biomarker for closed-loop DBS."

I would recommend re-wording this statement. "Real-life" seems unnecessary. Perhaps "electrophysiological"? Or do authors mean "real-time"? There is an understandable tendency by many authors to claim whatever physiological feature they identify in their study as being useful for closed-loop DBS. I would encourage the authors to be a little more specific in this statement about in what way it would be useful for closed-loop DBS. To modify stimulation amplitude? To impact when to stimulate? To alter contacts to stimulate on? Universal or limited to those with FTG? Extensive detail isn't warranted in the abstract, but a little more specificity would be beneficial.

Intro

"DBS-induced entrainment was identified in a few patients at 1/2 of the DBS frequency while on or off dopaminergic medication.^{7,8}"

While citation 7 reported 1:2 entrainment, citation 8 did not find that shifting of DBS frequency changed the frequency of FTG. In fact the authors of citation 8 state in their discussion: "the evidence argues against entrainment of the FTG by DBS off medication". This sentence should be revised to accurately reflect the findings of these studies.

Methods

"Contact selection in the remaining 10 patients (SenSight electrodes) was based on the occurrence of gamma entrainment during a monopolar review in OFF-medication, which is described elsewhere.⁴⁰" The cited study pertains to monopolar review and beta oscillations, but not gamma entrainment. Can the authors describe contact and amplitude selection when the targeted FTG physiological marker was not found? (4 patients)

For completeness it would be useful to include in the table a third category of UPDRS-III, on meds + on stim, if collected.

"LFPs were recorded during a systematic unilateral ramping of stimulation amplitude at rest" Example of this looks like the spectrograms in Figure 1. Was this done only once per patient, or was the ramping protocol repeated to determine repeatability of results?

Since this ramping protocol was performed in all subjects, I think spectrograms shown in Figure 1 should be shown for all patients as a supplementary figure, so readers get a clearer sense of the types and diversity of responses that are observed.

Results

It would be preferable for spectrograms to include a colormap

Supplementary figure would be a little easier to understand if the related spectrogram for that patient was included (like shown in Figure 1a)

“Gamma entrainment was restricted to the STN that was stimulated and did not transfer to the non-stimulated other hemisphere in any of our cases.” That patients were implanted (and recorded from) bilaterally was not described I do not think in the methods.

Figure 2B the inset figure makes this figure panel cluttered and more confusing. (also considering different x-axes) It is not immediately clear what it represents or what the purpose is of oval outline light grey box around time 60sec

“revealing a tendency of increased intermediate activity before entrainment.” It is not readily apparent that this is the case, that intermediate activity is increased before entrainment compared to earlier or later in the recording. Could this not be quantified and statistically compared?

Figure 3 shows group comparisons with vs without FTG. However as shown earlier, different levels of stimulation can produce different levels of FTG. Could data be analyzed to examine the potential correlation between magnitude of entrained FTG and magnitude of motor improvement? The title of Figure 3 “DBS-induced entrainment is modulated by movement and improves motor performance.” modulation by movement is evident however as worded it appears to claim a causal relationship between FTG entrainment and improvement in motor performance which is a little bit of a stretch.

Discussion

Authors state “the peak frequency of spontaneous FTG gradually decreased until it locked to $\frac{1}{2}$ DBS frequency,” though as noted above this claim should be more clearly supported by the data.

“This observation fits the hypothesis that entrainment depends on the presence of endogenous oscillatory activity belonging to physiological processes”, how do the authors explain the presence of DBS induced FTG in patients without apparent endogenous off-stim FTG?

“Thus, DBS-induced entrainment may enhance (patho)-physiological oscillatory activity, and this may explain part of the prokinetic effect of subthalamic high-frequency DBS and its effectiveness in alleviating bradykinesia.” Are the authors referring to low frequency DBS entrainment or now back to HF DBS? If HF DBS, rather than enhance I believe it should say suppress.

Can authors comment on stimulation delivered in the gamma range (60-80Hz), which a number of studies have explored? Based on this entrainment hypothesis, should we expect that stimulation delivered at gamma band frequencies to produce therapeutic effects by enhancing prokinetic gamma signals? Or is there something special about the 1:2 entrainment that produces greater effects than 1:1 entrainment?

“Our data suggest that this biomarker is not solely associated with dyskinesia but could be assigned to a prokinetic state that should be maintained. “ There is high inter-subject variability with FTG, and some patient’s optimal clinical setting is lower than when FTG appears, others greater. Can authors comment on the potential patient specific characterization that would likely be required to employ adaptive algorithms that utilize FTG?

“We show that the amplitude of the entrainment increased during movement and that motor performance is improved with entrainment. “

I think this slightly overgeneralizes the findings. To me it a more accurate way to describe the results is that it appears that the amplitude of the entrainment increased during movement in patients with DBS-induced FTG, and that motor performance was better in that group than patients without DBS-induced FTG.

As mentioned earlier, if authors were able find correlations within subject and/or across subjects regarding level of entrainment relative to degree of motor improvement, that would provide stronger evidence to make this claim.

Reviewer #3

(Remarks to the Author)

This is a very interesting, well-written manuscript investigating half-harmonic entrainment in patients with Parkinson’s disease. The study boasts a relatively large sample size for this type of study and, crucially, relates half-harmonic entrainment to motor performance, thereby providing evidence of its clinical relevance.

The manuscript is to the point, and I only have minor comments.

-I would be grateful if the authors could add line numbers in future submissions (and refer to them when listing their changes)

-Introduction

o Ref 8 does not show 1:2 entrainment in the off-medication state. The signals measured do not occur at half the frequency of stimulation, so are not consistent with entrainment. Could the authors please clarify or remove the statement.

-Results

o Fig 1B: in a number of patients the stim amplitude was not increased much. 1:2 entrainment may have disappeared in these patients at higher stimulation amplitudes. I would suggest mentioning this in the limitation section.

o How do the authors determine whether entrainment is present or not in the data?

o Fig 2B: I suppose the authors are using the average modulus of the analytic signal, not the average analytic signal (which is a timeseries of complex numbers). This should be fixed in the figure/caption/main text.

o Given the clinical data in Table 1, the authors may be able to check if the clinical benefit of DBS was lower in patients who did not display entrainment?

o Bottom of page 7: LME abbreviation is undefined.

-Discussion

o Page 8: "that there are more, previously unexplored mechanisms" >> more is strange here, can simply be deleted.

o Page 9: "This observation fits the hypothesis that entrainment depends on the presence of endogenous oscillatory activity". The hypothesis that entrainment is more likely when endogenous oscillator activity is present was also supported by modelling in a recent letter (<https://doi.org/10.1016/j.brs.2024.02.017>).

o Page 9: "Entrainment caused by subthalamic DBS would typically appear in the gamma band around stimulation frequency (130 Hz) or its subharmonics (60-65 Hz)." >> the frequency ranges at 1:1 and 1:2 should be made consistent.

-Methods

o Contact selection is based on beta in 9 patients, but on gamma entrainment in the rest of the patients. In the former case, are patients less likely to show gamma entrainment?

o Stimulation is unilateral, how were the hemispheres used in the study selected?

o "Since these phases have varying durations, we resized each phase to a common duration by resampling the smoothed z-scored analytic signal. Resampling the z-scored analytical signals preserved the overall dynamics of the signal, while allowing for group level analyses." >> is this because the phases have varying durations across patients (rather within patients)?

o Page 15: "We further investigated the effects of varying DBS settings on three different gamma phenomena, i.e., FTG without stimulation..." >> the formulation implies that DBS would have an effect on FTG without stimulation, I would suggest to rephrase this.

Version 1:

Reviewer comments:

Reviewer #1

(Remarks to the Author)

The authors did a great job responding to my comments. Congratulations on this important work.

Reviewer #2

(Remarks to the Author)

Overall the authors provided very thorough and adequate response to reviewer comments. Changes and additions made have strengthened the manuscript.

One remaining point:

Regarding response to Reviewer 2 Comment 16, and discussion paragraph beginning with line 249, I do still think it would be useful to include some discussion about potential gamma entrainment with gamma range stimulation. At the end of their response to Comment 16, they state "we restrain from commenting [on] lower frequency stimulation in our manuscript." The authors state, however, in the discussion Line 254 that "...This hypothesis is supported by previous studies that showed that stimulation at 20 Hz (i.e. within the beta band of 13-35 Hz) worsened motor performance and bradykinesia. In this case, entrainment possibly promoted activity within the beta frequency range that is considered pathological and enhanced parkinsonian symptomatology."

Given what the authors described as likely entrainment in the beta band range during beta band (20hz) stimulation (I presume they are considering likely producing 1:1 entrainment), which they state supports their hypothesis that "entrainment per se could facilitate the physiological mechanism that is related to the 1:2 entrainment frequency, in this case likely the prokinetic gamma synchronization," would it not also make sense to assume that gamma band (1:1) entrainment is likely occurring during stimulation delivered at gamma band frequencies as performed in previous studies? Are those studies not relevant to the entrainment hypothesis, similar to the 20hz stimulation studies? It seems to me an interesting unanswered question to raise in this discussion section why motor signs like bradykinesia would not be particularly improved during

gamma range stimulation. Based on the entrainment hypothesis that gamma synchronization is prokinetic, readers might think it would make sense to stimulate and entrain at gamma frequencies, though as the authors noted in their response from a clinical perspective this is not very effective. It is my opinion that some additional discussion of this topic would be warranted in this section.

Reviewer #3

(Remarks to the Author)

I would like to thank the authors for having appropriately addressed my comments. Two minor points:

- 1) The more common terminology is the analytic signal, not the analytical signal.
- 2) It would be good to caveat the statement added Lines 103-108 by mentioning that the test was underpowered (n=4 in group without entrainment).

POINT-BY-POINT RESPONSE TO REVIEW

Reviewer #1 (Remarks to the Author):

The authors investigate the role of DBS-induced gamma entrainment as a potential biomarker for motor improvement in Parkinson's disease (PD). They report that DBS at therapeutic levels entrains gamma oscillations in the STN at half the stimulation frequency in a majority of patients, which was associated with enhanced motor performance but not necessarily dyskinesia. The results suggest that DBS-induced gamma entrainment as a candidate biomarker for adaptive DBS, with implications for optimizing motor outcomes in PD patients.

1. To me the most interesting aspect of this paper is the modulation/interruption or creation of gamma by DBS. What however does 'finely tuned' gamma mean? It seems to place a kind of value judgement on an oscillation that is incompletely defined. What makes it 'fine'? What makes it 'tuned' (I supposed the association with the DBS frequency, but in what way is it tuned when it occurs spontaneously?). Prior work suggesting this is a biomarker for unwanted dyskinesias would not in my mind make it 'fine,' maybe this refers to it being narrow- rather than broad-band?

We thank the reviewer for the constructive comments. This is an important nomenclature concept highlighted. Gamma rhythms, typically ranging from 40 to 100 Hz or higher, are commonly observed throughout the human motor system. Current hypotheses suggest that gamma activity is a rather heterogeneous and broad band, comprising distinct sub-bands that serve as markers for different behavioral or clinical characteristics. We try to distinguish 3 different phenomena in the gamma band:

Finely-tuned gamma (FTG) is indeed the narrowband-activity which is typically encountered between 70-90 Hz both in cortical and subcortical regions. FTG has been prominently studied in patients with Parkinson's disease, particularly in the on-medication state and has been linked to levodopa-induced dyskinesia.^{1,2} FTG occurs spontaneously, and we use this term for the spontaneous narrowband gamma activity in line with the literature.^{3,4} This is to differentiate it from stimulation-induced gamma activity that we describe in our patients during DBS. In the latter case we used the term *DBS-induced gamma entrainment*. This was selected (rather than DBS-induced FTG) to describe the phenomenon by which neuronal populations synchronize/entrain in response to an external stimulus, here, electrical stimulation. Finally, *broad-band gamma* is an event-related synchronization phenomenon, typically elicited during voluntary movement also in cortical and subcortical regions. Current literature has shown that this type of synchronization is associated with successful motor execution.

These distinctions are becoming increasingly important in the context of DBS research and the treatment of patients with PD. To clarify this distinction, we add the following to the Introduction of our manuscript (page 3: 'Introduction'; added statements in italics underlined):

'...Recently, LFPs recorded during high-frequency DBS in patients with PD unveiled an entrained neural signal within the gamma band (40-100 Hz) in both cortical and subcortical recordings. *DBS-induced gamma entrainment was identified in a few patients as a narrow-band activity with a peak at ½ of the DBS frequency during the on-dopaminergic state.*¹ Based on model predictions, 1:2 entrainment is hypothesized to appear only in certain combinations of DBS amplitude and frequency^{5,6} however, its functional role is largely unexplored yet. *Broad band gamma activity (40-100 Hz) in subcortical or cortical LFPs* has been closely linked to the initiation and execution of movements and is considered to have a prokinetic effect.⁷ In PD patients, a more narrow frequency band (70-90 Hz), referred to as spontaneous finely tuned gamma (FTG), is associated with levodopa-induced dyskinesia (LID), a condition in which PD patients experience involuntary movements as a result of long-term treatment with dopaminergic medication.^{1,8,9} *Here, we use the term 'spontaneous' to indicate that this activity*

occurs at rest, in contrast to the movement-related broad band gamma ERS. Moreover, it is different from the DBS entrained neural activity. Neural entrainment describes the phenomenon by which ongoing brain oscillations synchronize to the rhythmic pattern of an external stimulus, such as repetitive auditory cues, visual signals, or electrical stimulation.¹⁰⁻¹⁴ Successful neural entrainment by an external stimulus enhanced the cognitive or motor function that was associated to the oscillatory activity being entrained.^{15,16'}

2. The authors provide frequency domain data for neural entrainment by DBS but neglect decades of prior research on DBS-evoked potentials recorded from various sites. Event related potentials measure neural synchronization, in this case to the DBS pulse. The prior work indicates that at some level DBS entrains scalp potentials over ipsilateral motor cortex (Baker KB et al 2002), including at therapeutic high frequencies (Walker HC et al 2012). Recent work also provides evidence for entrainment within both the STN (Sinclair N et al 2018) and the GPi (Awad et al 2021, Johnson K et al 2024).

We agree with the reviewer that DBS can produce manifold evoked responses which can be detected both cortically^{17,18} and subcortically.^{11,19} Cortical responses have been associated to the hyperdirect pathway²⁰ while subcortically recorded evoked responses, namely evoked resonant neural activity (ERNA), have been suggested to be elicited by synaptic dynamics within the indirect pathway.²¹

In this study, we focus on the recently described gamma entrainment that is not well understood yet. DBS at high frequency (130 Hz) evokes a large-amplitude resonant neural activity (ERNA) to each DBS pulse. ERNA is more prominent in the dorsal part of the STN and has been linked to good motor outcome. Recently, ERNA was also recorded in the GPi during high-frequency DBS and was found to be more prominent in the postero-dorsal part of the pallidum. In most cases, the contacts where ERNA was higher in amplitude were later used as optimal sites for chronic therapeutic stimulation.²² In the present report, subcortically evoked oscillations have been studied in the chronic setting with an IPG that is constrained in sampling rate and thus does not allow for the recording of ERNA which is of frequencies above 300 Hz. The described 1:2 entrainment is less often observed in externalized recordings where ERNA is visible (performed with devices with higher sampling rate), likely due to the stun effect. We speculate that both entrainment and ERNA are based on circuits relevant to motor improvement. However, with the currently available IPGs we are not able to disentangle the potential relation between these phenomena. Future generations of IPGs may allow for such recordings and thus the more in-depth analysis of the relation between these signals, thus potentially allowing for further inferences about their network origin.

Evoked potentials and ERNA offer important information about network states induced by DBS, and therefore this literature has been added to our 'Introduction'.

The following passage has been added to the Introduction (page 3-4, **Lines 49-53**):

'Additionally, DBS can induce evoked responses which can be detected both cortically^{17,18} and subcortically.^{11,19} Cortical responses have been associated to the hyperdirect pathway²⁰ while subcortically recorded evoked responses, namely evoked resonant neural activity (ERNA), have been suggested to be elicited by synaptic dynamics within the indirect pathway²¹.

3. The one-half frequency analyses do not evaluate for entrainment at the stimulation frequency, likely because of the large electrical stimulus artifact (~130 Hz is truncated from the y axis). So while interesting in its own right, my suspicion is that this half frequency phenomenon is a network behavior that can become superimposed over entrainment at the stimulation frequency. Single pulses elicit complex waveforms are longer than the interstimulus interval during therapeutic

stimulation. As such, the prior stimulation history would likely induce summation effects that could amplify or suppress gamma in complex ways that depend on patient-specific network characteristics, DBS electrode location, stimulation parameters, and other factors. I think the current work would benefit from showing evoked potential waveforms to the stimulus pulse in these patients to get a better sense of how stimulation is driving the network in the time domain.

- One notable strength is that this work is performed with a novel device with recording and stimulation capabilities in chronically implanted patients.

- In any case, DBS-induced gamma entrainment could serve as a novel biomarker for motor improvement, which could provide a parameter to guide closed-loop DBS strategies. This work demonstrates potential new avenues for improving motor performance in PD patients.

- Experimental design and setup is clear with a focus on the relationship between gamma entrainment and motor performance.

This is an interesting speculation on the origin of the $\frac{1}{2}$ DBS frequency phenomenon that we will consider further. In this study, our LFP data are recorded with the PERCEPT PC IPG, with a sampling frequency of 250 Hz. Since all STNs are stimulated with 130/125 Hz (3389/SenSight electrodes respectively), there is indeed a large stimulation artefact in this frequency. Additionally, there is an internal hardware filter of the DBS pulses, and these are not visible in the raw LFP data when extracted from the recording device. This makes it impossible to show evoked potential waveforms to the stimulus pulse in the time domain. We can only estimate that with a DBS frequency of 125 Hz, a pulsewidth of 60 μ s, and a sampling frequency of 250 Hz, there is one DBS pulse every 0.008 seconds, or every 2 samples. **Supplementary Figure 1** of the original manuscript shows LFP data filtered around the peak frequencies of interest (FTG and $\frac{1}{2}$ entrainment) with estimated DBS pulses. Externalized recordings, when the DBS electrodes have not been connected to the IPG yet offer the possibility to record LFPs with higher sampling rate of e.g. 4 kHz, and the DBS pulses visible to the raw data. This would facilitate the analysis of evoked potentials. Unfortunately, our experience has shown that $\frac{1}{2}$ entrainment is often not seen in externalized recordings, potentially due to the stun effect. In a separate study, we will follow up this hypothesis in our externalized recordings if possible. Within this framework, it will be important to elucidate the transition period from spontaneous FTG to entrained gamma that we observed in our patients and that strengthens our assumption of an entrained oscillation. In the same vein, in a subset of patients the power of 1:2 entrainment was reduced with higher DBS amplitude.

The current study, however, aims to explore entrained gamma activity as a potential new biomarker for adaptive DBS in a large cohort of patients using the newly available IPG. We have now explained in the methods and in Suppl. Fig. 1 the limitation of the hardware filter. Moreover, we have included the hypothesis on the potential origin of 1:2 entrainment regarding evoked activity to the discussion.

The following changes have been applied to the manuscript:

Page 16: 'Methods', **Lines 403-405:**

'The stimulation pulses were internally filtered by the hardware, therefore their timing could only be estimated based on the DBS frequency and pulsewidth'.

Supplementary Figure 1 – Caption:

'The recovery of the actual DBS pulses was not possible due to hardware constrains. DBS was applied with a frequency of 125 Hz and pulse width of 60 μ s, therefore estimated DBS pulses are plotted every

8 ms. During 1:2 entrainment it is visible that a peak of activity appears every two DBS pulses. Waveforms of evoked responses could not be recorded due to the hardware constraints'.

Page 10: 'Discussion', Lines 221-225:

'Evoked potentials¹⁸ and resonant neuronal activity²¹ that have been described in the STN-GPi loop have to be considered as also causing entrained gamma activity. However, constraints of the hardware filters when using Percept limit any further analysis of high frequency and evoked activity'.

4. A substantial number of patients did not display the gamma entrainment phenomenon (n=4); presumably these patients improved with DBS. What were their motor improvements? What is the improvement in contralateral UPDRS part 3 sub-scores in patients who did versus did not exhibit entrainment? Is the one half gamma entrainment necessary for improvement.

It is correct that all patients showed a significant improvement with DBS. We consider 1:2 gamma entrainment a potential state biomarker that relates to good motor states as shown by the motor performance in our patients. Specifically, we show that the subgroup that exhibited 1:2 gamma entrainment performed better than the group without entrainment in our motor task, a difference that was not influenced by the DBS intensity. This motor parameter was assessed at the time of the entrainment. This is also in line with the recent first application of entrained gamma activity as a feedback signal for aDBS in 4 patients.²³ In contrast, full UPDRS III scores were assessed during standard conditions on the ward not in parallel with the Percept recording and potential fluctuations in patient condition between the time of recording and UPDRS assessment have to be considered. A direct comparison between UPDRS scores in patients with and without entrained gamma recorded at a later time point is very limited and has not shown a difference between groups. Irrespective of this, we do not think that entrained gamma activity is the only mechanism of DBS necessary for successful motor improvement. It might be a phenomenon that occurs at a specific level of (levodopa-induced) improvement potentially near the threshold for dyskinesia. We also have described that in some patients higher stimulation amplitude leads to disappearance of entrainment, in line with an optimal parameter space for entrainment defined by the Arnold's tongue. This might lead to a better motor response but is not necessarily a prerequisite for the DBS being effective at all.

According to the reviewer's suggestions, we performed comparisons of motor improvement (expressed as percentage change of MedOn-StimOn to MedOn-StimOff) in the clinical scores (UPDRS-III) of patients that exhibited vs the patients that did not exhibit DBS-induced 1:2 gamma entrainment. The two subgroups had substantially different number of patients (N=15 with entrainment vs N=4 without entrainment), therefore the second sample is largely underpowered for meaningful statistical inferences. For one patient UPDRS-III score was not available (subgroup without entrainment; final N=3) No statistically significant differences were found, showing that all the patients (regardless of the gamma activity being present at a later time point) improved with DBS.

The following has been added to the manuscript (page 5-6: 'Results', Lines 103-108):

'UPDRS III motor score improved in all patients with DBS (MedOn-DBSOff $\mu = 29.72 \pm 13.34$, MedOn=DBSON $\mu = 17.47 \pm 10.07$, $P = 0.007$). With respect to entrainment, there was no significant difference in the On-medication motor improvement with DBS using standard clinical amplitude settings between the subgroup of patients that exhibited entrainment and the subgroup without entrainment both in total scores, as well as hemibody subscores'.

5. They should correlate the amount of entrainment with the amount of improvement on their motor task.

We thank the reviewer for this advice. We have assessed a linear correlation (Spearman Rho) between motor improvement during the repetitive finger tapping task, as measured by the relative change in four movement metrics compared to DBS-Off, and the absolute entrainment power during movement (**Figure 1A**), as well as the correlation between entrainment power during movement relative to rest (**Figure 1B**). Additionally, we report the results of a Linear Mixed Effects (LME) model, which examines the relationship between absolute entrainment power during movement and motor improvement across all blocks of finger tapping, accounting for repeated measures (**Figure 1C**). Neither the correlations nor the LME model revealed statistically significant results.

A potential explanation for the lack of a linear relationship between entrainment power and motor improvement is that entrainment power may not increase in a continuous fashion but rather in a stepwise manner and within a defined parameter space (according to the Arnold's tongue model) This stepwise increase, induced by DBS, could be what supports successful motor performance. Despite the absence of a direct correlation, there was a significant difference in motor performance between the group that had entrainment and the group that did not, specifically in two movement metrics (mean raise velocity and the coefficient of variation in intra-tap intervals). Importantly, DBS amplitude did not emerge as a significant confounding factor in these differences (See Figure 3C in the original manuscript). Thus, occurrence of entrained gamma activity could still serve as a state biomarker for improved motor performance.

We add the following statement to the manuscript (Page 8: 'Results', **Lines 173-176**):

'There was no significant correlation between the power of entrainment during movement and the improvement in the motor task on the above metrics, pointing to a step-wise increase in entrained gamma activity with movement'

We also re-phrased the following sentences in our discussion to tune down our statement of motor improvement and entrained gamma activity (see Reviewer 2) (Page 11: 'Discussion', **Lines 249-254**)

"More importantly, we show that patients that exhibited DBS-induced gamma entrainment performed better in the finger tapping task than the patients that did not. Although entrainment amplitude did not correlate with motor improvement, we hypothesize that entrainment irrespective of its amplitude could enhance the physiological mechanism that is related to the 1:2 entrainment frequency, in this case likely the prokinetic gamma synchronization".

and Page 10: 'Discussion' title, **Lines 227-228**:

'DBS-induced entrainment is modulated by movement and is present in patients that had better performance in a motor task'

Figure 1. Relationship between entrainment power and motor improvement during a repetitive finger tapping task. (A) Correlation between absolute entrainment power during movement and motor performance relative to DBS-off condition. **(B)** Correlation between entrainment power during movement relative to rest and motor performance relative to DBS-off. **(C)** Linear Mixed Effects (LME) model assessing the impact of DBS-induced 1:2 entrainment on movement metrics across all repetitive movement blocks, with motor performance relative to DBS-off. The model accounts for repeated movement blocks.

6. These are on medication studies, so a component of the motor performance likely relates to how dopaminergic medications impact individual participants, not related to stimulation alone. The gold

standard for understanding motor improvements related to DBS is off dopaminergic medications >12 hours.

Indeed, the extent of the effect of DBS on the improvement of parkinsonian symptoms is best studied in the off-dopaminergic state, where the patient-specific response to dopaminergic treatment is not a confounding factor. However, in clinical routine motor improvement is achieved by a combination of both DBS and dopaminergic medication that have additive effects. The additional effect of DBS even in the On-medication state was also proven in our patients (Med On-DBS Off $\mu = 29.72 \pm 13.34$, Med On=DBS On $\mu = 17.47 \pm 10.07$, $p = 0.007$). Interestingly, so far DBS-induced 1:2 gamma entrainment is mainly described in the On-medication state. A reason for the limited evidence of entrainment in the off medication condition could be that the entrained signal is more prominent when an endogenous ongoing oscillatory activity pre-exists, here spontaneous FTG.²⁴ A further hint towards this assumption is that the DBS amplitude needed to induce gamma entrainment was smaller in those patients with FTG, however, numbers are too small and statistical comparison was not significant.

Finally, conducting studies in the on-medication state is crucial, as it more accurately reflects their real-life conditions compared to off-medication state. This is especially important for the development of effective closed-loop DBS systems.

Nonetheless, it would be interesting to see whether DBS-induced entrainment is also visible in the off-medication state and whether it has similar characteristics and correlates as the signal in the on-dopaminergic state in the context of PD. Please see our response to question 8 that further evolves on this topic.

7. The study focuses on short-term symptomatic effects, limiting insights into the long-term implications of gamma entrainment on motor function and dyskinesia.

Indeed, we have not analyzed the long term effects in this first large study on gamma entrainment. The current study was performed in the laboratory environment, with two important clinical readouts (UPDR-III score and dyskinesia occurrence as assessed by a medical doctor), as well as a classic motor task of repetitive finger tapping, assessing bradykinesia. However, it does not offer insights into the long-term correlation of DBS-induced gamma entrainment and motor performance/clinical characteristics. The goal of the current study was to characterize the 1:2 entrainment, to establish whether it is merely an artefact and to investigate its short-term modulation. The advantage of our study is the relatively large patient cohort that has been carefully examined under defined lab conditions. As a next step, long term modulation could be studied in real life condition. Here, it has to be considered that using Percept for chronic recordings, for now only mean activity of 5 Hz around the peak frequency of interest (here, $\frac{1}{2}$ of the DBS frequency) can be collected every 10 minutes as an average power value. This would provide interesting additional data on circadian fluctuations and can be aligned with clinical motor fluctuations over time but has not the fine-grained resolution of our current data that provide a more detailed insight of frequency modulation and time points of entrainment. Importantly, a first proof-of principle clinical application in 4 patients (using a different device that is only available for a limited number of subjects in the US) has shown that entrained gamma is a potential biomarker for closed-loop DBS. A follow-up study with this cohort could involve measuring chronic entrainment using the chronic sensing function in percept and wearables for dyskinesia assessment. We have added this to the discussion (see point 8).

8. The study does not systematically vary dopamine levels, which could further clarify the interaction between dopaminergic medication and gamma entrainment and how this might relate to a closed loop DBS paradigm.

On a similar note, in the current study we do not systematically vary dopamine levels, rather all patients received 100-200 mg of fast-acting levodopa. A future study taking this into account could unveil a more refined interaction of levodopa-induced FTG and DBS-induced gamma entrainment with increasing medication intake. From the current study we hypothesize that dopamine levels will have a major influence on these signals together with other factors not investigated here, such as the location of the selected contact for DBS administration. We add this limitation to our 'Discussion' (Page 13, Lines 319-325):

'Future studies could focus on recording 1:2 gamma entrainment chronically, while patients are at home and experience all different levels of dopaminergic medication intake, together with other daily activities and dyskinesia severity. This approach could help optimize its potential for use amplitude modulation in closed-loop DBS systems. Additionally, the influence of contact selection for stimulation could be investigated to disentangle other DBS parameters that influence 1:2 entrainment'.

9. In multiple places, the authors say something is changed versus something else, but that it did not reach statistical significance. My view is we have to go by the statistics. It is either significant or not significant at their level of statistical power. As written, they read as interpretations. I don't think the inferences are justified without statistical significance, although they might offer some speculations for future hypothesis testing in the discussion.

This is a valid point by the reviewer. When the result is not statistically significant, there is insufficient evidence to support any difference between the compared groups, leading to overinterpretation. The differences that were not statistically significant are the ones below:

1. The DBS amplitude in which 1:2 entrainment was present was 1.91 ± 0.68 mA in patients with FTG and 2.6 ± 0.62 mA in patients without spontaneous FTG, which was not significantly different ($P = 0.162$). We agree that further statistical evidence is needed to speculate on potential influence of levodopa. This paragraph is therefore reframed in the manuscript as following (Page 5: 'Results', Lines 93-95):

'The patients showing spontaneous FTG developed 1:2 entrainment at a mean DBS amplitude of 1.91 ± 0.68 mA and the patients without spontaneous FTG at $\mu = 2.6 \pm 0.62$ mA, which was not significantly different ($P = 0.162$)'.

2. The power of the entrained gamma was smaller when STNs were stimulated with higher DBS frequency. This comparison is between 4 patients, a sample size that is substantially small to have any statistical support. Nevertheless, this is a hypothesis-driven observation that fits the previously validated framework of Arnold Tongues.⁵ We describe the observation and discuss it briefly in the light of the Arnold Tongues, which could be valuable for future studies.

'However, entrained gamma power had smaller values with higher stimulation frequency, although the small sample size did not allow for a statistical analysis (110 Hz $\mu = 0.08 \pm 0.06$ μ Vp; 125 Hz $\mu = 0.03 \pm 0.005$ μ Vp, 145 Hz $\mu = 0.02, \pm 0.003$ μ Vp)'.

3. When comparing the motor performance between the group that showed entrainment (N=9) and the group that did not (N=4), 2/4 metrics, were significant, the Normalized Root Mean Square of the trace showed a trend ($p=0.058$) and the Coefficient of Variation of the Impact Force was not statistically different between the two groups. This is now rephrased as follows: (Page 7-8: Results, Lines 169-171):

'Normalized RMS of the trace shows a trend ($P = 0.058$, CI: $-0.007-0.123$) and coefficient of variation of the impact force did not yield into statistical significance ($P = 0.184$, CI: $-0.126-0.024$)'.

Reviewer #2 (Remarks to the Author):

The authors present data on a topic that is of increasing interest to researchers in the field of DBS. Although adaptive DBS approaches are hoped to provide a way to improve patient outcomes with DBS, a better understanding of potential biomarker signals to use in control algorithms is essential. This study adds to our understanding of one such biomarker signal: finely tuned gamma oscillations. The manuscript is generally well written. For the most part data is presented that supports the authors claims, but there are several instances where I believe the authors overstate their findings as indicated in comments below. The claim that FTG entrainment improves motor performance could be reworded or be better supported by a correlation analysis directly relating the amount of entrainment to the degree of motor improvement.

We thank the reviewer for their constructive feedback and for acknowledging the timeliness of our study, particularly in the context of closed-loop DBS. Below, we provide a point-by-point response and outline the changes made to the manuscript, aimed at clarifying our findings where there may have been a risk of overstating them.

Specific comments according to manuscript section are below.

Abstract

1. "DBS-induced entrainment can be a promising real-life biomarker for closed-loop DBS."

I would recommend re-wording this statement. "Real-life" seems unnecessary. Perhaps "electrophysiological"? Or do authors mean "real-time"? There is an understandable tendency by many authors to claim whatever physiological feature they identify in their study as being useful for closed-loop DBS. I would encourage the authors to be a little more specific in this statement about in what way it would be useful for closed-loop DBS. To modify stimulation amplitude? To impact when to stimulate? To alter contacts to stimulate on? Universal or limited to those with FTG? Extensive detail isn't warranted in the abstract, but a little more specificity would be beneficial.

We thank the reviewer for the chance to clarify how we hypothesize DBS-induced 1:2 gamma entrainment could serve as a biomarker for closed-loop DBS. In this study, we demonstrate that this signal is associated with better motor performance in the dopaminergic "on" state, while it is not consistently linked to levodopa-induced dyskinesia. Therefore, we propose that it could serve as a valuable marker for maintaining DBS and possibly titrating the amplitude in response to the entrained activity. Specifically, in the absence of this signal, increasing the DBS amplitude may be beneficial. These hypotheses are described in more detail in our 'Discussion' section, where we make some speculations about the relationship between FTG/entrainment and dyskinesia (sub-chapter 'Finely tuned gamma activity in the motor network and dyskinesia'), as well as about the utility of this signal for closed-loop DBS (subchapter 'DBS-induced gamma entrainment and closed-loop DBS'). We modify our abstract as follows to make the final statement more precise (Page 2: **Lines 15-17**):

'DBS-induced entrainment can be a promising neurophysiological biomarker for identifying the optimal amplitude during closed-loop DBS'.

Additionally, we added this specific hypothesis to the discussion as follows (Page 13, **Lines 320-323**):

'Future studies could focus on recording 1:2 gamma entrainment chronically, while patients are at home and experience all different levels of dopaminergic medication intake, together with other daily activities and dyskinesia severity. This approach could help optimize its potential for use amplitude modulation in closed-loop DBS systems'.

Intro

2. "DBS-induced entrainment was identified in a few patients at ½ of the DBS frequency while on or off dopaminergic medication.7,8" While citation 7 reported 1:2 entrainment, citation 8 did not find that shifting of DBS frequency changed the frequency of FTG. In fact the authors of citation 8 state in their discussion: "the evidence argues against entrainment of the FTG by DBS off medication". This sentence should be revised to accurately reflect the findings of these studies.

Citation #8 of the original manuscript refers to Wiest et al., 2021 'Subthalamic deep brain stimulation induces finely-tuned gamma oscillations in the absence of levodopa' (*Neurobiology of Disease*). This study reveals narrow-band activity between ~70 to 90 Hz during continuous stimulation in the off-dopaminergic state. The motivation for this citation was not driven by this result, rather the following statements of this study, which revealed that there was a subharmonic in the ½ DBS frequency present in this cohort:

Citation from Wiest et al.: Wiest C, Tinkhauser G, Pogosyan A, He S, Baig F, Morgante F, Mostofi A, Pereira EA, Tan H, Brown P, Torrecillos F. Subthalamic deep brain stimulation induces finely-tuned gamma oscillations in the absence of levodopa. *Neurobiology of disease*. 2021 May 1;152:105287.

"2.6. Signal processing: pre-processing

'These artefacts include line noise at 50 Hz (all recordings), a peak at the subharmonic frequency of 65 Hz during stimulation at 130 Hz in 11/17 STNs (and all 5 patients displaying FTG)'.

2.7 Signal processing: FTG detection

We restricted our search for the FTG to the frequency range from 65 to 90 Hz (Kempf et al., 2009; Swann et al., 2016) to avoid DBS sub harmonics'

Similarly, in Figure 1C of the paper, one can detect the 1:2 subharmonic (label 4) and the corresponding figure caption: 'DBS-induced narrowband artefacts at a subharmonic DBS frequency (65 Hz; label 4) or due to an aliasing phenomenon (80 Hz; label 5)'."

In this study this signal is treated as artefactual (4.1.: 'Totally discounting a physiological contribution to those subharmonics is difficult, although their narrow, fixed frequency and their duration precisely matching the duration of stimulation might point more to stimulus artefact').

However, we agree that our statement that DBS-induced entrainment was identified in the off-medication statement is misleading, as in the cited study there is no previous ongoing oscillatory activity that is being entrained into a subharmonic. To prevent any confusion, we removed this citation from our introduction (Page 3: **Lines 32-34**).

*'DBS-induced entrainment was identified in a few patients as a narrow-band activity with a peak at ½ of the DBS frequency during the on-dopaminergic state.'*¹

Methods

3. "Contact selection in the remaining 10 patients (SenSight electrodes) was based on the occurrence of gamma entrainment during a monopolar review in OFF-medication, which is

described elsewhere.⁴⁰ The cited study pertains to monopolar review and beta oscillations, but not gamma entrainment. Can the authors describe contact and amplitude selection when the targeted FTG physiological marker was not found? (4 patients).

From the cohort included in the current study all SenSight patients (N=10) were recruited (hemisphere & contact) based on highest gamma. All 3389 patients (N=9) were retrospectively added and were part of previously published studies.^{25,26} These were selected based on the contact with highest beta. A previous study presented a computational model which demonstrates that the contact with the higher beta band power is similar to the one with the highest gamma band activity.²⁷ Since due to time constrains it is rather challenging to test all different contacts for the optimal gamma activity, we opted for the contact with the highest beta in these retrospectively added patients.

This is now clarified in the Method section (Page 15, **Lines 385-387**):

'In these cases, contact selection was based on the strongest beta activity (13-35 Hz) during OFF-medication and other ring contacts were not tested'.

4. For completeness it would be useful to include in the table a third category of UPDRS-III, on meds + on stim, if collected.

We added the corresponding column (UPDRS-III Med On-Stim On) in Table 1 as follows:

Table 1. Demographic and clinical details

Patient	UPDRS-III Med Off- Stim Off	UPDRS-III Med On- Stim Off	UPDRS-III Med On- Stim On
#1	22	5	5
#2	78	34	23
#3	51	28	22
#4	39	11	11
#5	44	17	11
#6	43	30	7
#7	19	15	9
#8	53	43	12
#9	54	44	33
#10	49	39	31
#11	29	19	12
#12	51	39	21
#13	34	27	23
#14	57	35	11
#15	42	35	27
#16	42	26	14
#17	40	28	10
#18	38	-	9
#19	67	60	41

5. “LFPs were recorded during a systematic unilateral ramping of stimulation amplitude at rest”
Example of this looks like the spectrograms in Figure 1. Was this done only once per patient, or was the ramping protocol repeated to determine repeatability of results? Since this ramping protocol was performed in all subjects, I think spectrograms shown in Figure 1 should be shown for all patients as a supplementary figure, so readers get a clearer sense of the types and diversity of responses that are observed.

The systematic ramping of stimulation was performed once per patient in most of the patients in the current cohort. In a subset of patients (N=6) this was repeated twice, in which stimulation was administered in two different segments. However, this distinction is not part of the current analysis. The goal of the study protocol was to resemble a ‘monopolar-review’ paradigm, when increasing stimulation is used until the patient reports some notable side-effects. On that note, repeating the protocol multiple times would increase the study duration considerably, therefore patient fatigue would be a confounding factor.

A Supplementary figure (Supplementary Figure 2) has been added that includes all spectrograms recorded with the ramping stimulation amplitude in the group with FTG during DBS-off (A), without FTG during DBS-off (B), and without gamma activity (C). The numbers of the STNs correspond to the numbers throughout the manuscript, and the clinical details of the respective patient are presented in Table 1 of the manuscript. The new supplementary Figure 2 is provided below, and the corresponding reference has been added to Page 16: Methods (**Line 407**):

‘Individual spectrograms of all patients are presented in Supplementary Figure 2’.

(A) Group With FTG DBS-Off (STNs #1 - #8)

(B) Group Without FTG DBS-Off (STNs #9 - #15)

(C) Group with no Gamma Activity (STNs #16 - #19)

Figure 2. Individual spectrograms of all STNs. (A) STNs with spontaneous FTG during DBS-off. (B) STNs without spontaneous FTG during DBS-off. Note that figure of the STN #14 has a different color scale. (C) STNs without gamma activity detected on/off DBS.

Results

6. It would be preferable for spectrograms to include a colormap

Colormaps have been added to Figures 1A, 2A, and 3B of the original manuscript, as well as at the supplementary figures:

Figure 1A

Figure 2A

Figure 3B

7. Supplementary figure would be a little easier to understand if the related spectrogram for that patient was included (like shown in Figure 1a)

The respective spectrogram of the Raw LFP presented in Supplementary Figure 1, is in Supplementary Figure 2A, STN #5. This is now mentioned in the figure legend:

'The respective spectrogram for this patient is presented in Supplementary Figure 2A, STN #5'.

8. "Gamma entrainment was restricted to the STN that was stimulated and did not transfer to the non-stimulated other hemisphere in any of our cases." That patients were implanted (and recorded from) bilaterally was not described I do not think in the methods.

The following passages have been changed in the manuscript in Page 15; 'Methods':

Line 372: *'The patients were implanted bilaterally with Medtronic 3389 (n = 9) or SenSight (n = 10) DBS electrodes'.*

Lines 382-383: 'All sessions were video recorded. LFPs were recorded bilaterally in a bipolar montage between the two contact-rings adjacent to the stimulation level'.

9. Figure 2B the inset figure makes this figure panel cluttered and more confusing. (also considering different x-axes) It is not immediately clear what it represents or what the purpose is of oval outline light grey box around time 60sec.

The inset figure represents the average envelope of the analytic signal around the entrainment onset (point 0), with real time (not resampled) in the x-axis. Its goal was to show the dynamics of the gamma activity before entrainment onset, i.e. the increased intermediate activity in real time. To prevent confusion, the inset figure has been removed and Figure 2B of the manuscript is now as shown below:

10. "revealing a tendency of increased intermediate activity before entrainment." It is not readily apparent that this is the case, that intermediate activity is increased before entrainment compared to earlier or later in the recording. Could this not be quantified and statistically compared?

We statistically compared the average modulus of the analytic signal of the intermediate activity (as described in Methods). In the resampled data, the first 2500 samples (10 seconds) during DBS off, and the last 2500 samples before entrainment onset (with $\mu = 60.13\% \pm 23.09$ DBS intensity or $\mu = 1.51$ mA ± 0.58) were compared in a linear mixed effects model, while correcting for repeating measures. The two signals were significantly different ($p < 0.001$).

Page 7: 'Results' (Lines 142-143):

'With higher stimulation amplitude we observed a period when gamma activity was fluctuating within the intermediate frequency range (i.e. between spontaneous FTG and entrained activity) that continues until stable entrainment frequency is reached with further increase in stimulation amplitude. *This transition period is revealed by an increase of intermediate activity before entrainment in comparison to the DBS-off period ($P < 0.001$)*'.

Page 16: 'Methods' (Lines 428-430):

'To compare the fluctuation of these three signals, we extracted 2500 samples (10 seconds) during DBS-Off, 10 seconds after DBS was switched on, and the last 10 seconds before entrainment onset'.

11. Figure 3 shows group comparisons with vs without FTG. However as shown earlier, different levels of stimulation can produce different levels of FTG. Could data be analyzed to examine the potential correlation between magnitude of entrained FTG and magnitude of motor improvement?

We refer to the comment #5 from the First Reviewer for detailed tests and figures of the relationship between motor improvement and the power of 1:2 entrainment.

12. The title of Figure 3 “DBS-induced entrainment is modulated by movement and improves motor performance.” modulation by movement is evident however as worded it appears to claim a causal relationship between FTG entrainment and improvement in motor performance which is a little bit of a stretch.

Since no linear relationship between gamma entrainment and motor improvement has been established, we reframe the title as follows:

‘DBS-induced entrainment is modulated by movement and is present in patients that had better performance in a motor task’.

We also re-phrased the following sentences in our discussion to tune down our statement of motor improvement and entrained gamma activity (Page 11: ‘Discussion’, **Lines 249-254**).

“More importantly, we show that patients that exhibited DBS-induced gamma entrainment performed better in the finger tapping task than the patients that did not. Although entrainment amplitude did not correlate with motor improvement, we hypothesize that entrainment per se could enhance the physiological mechanism that is related to the 1:2 entrainment frequency, in this case likely the prokinetic gamma synchronization.”

and Page 10: ‘Discussion’ title, **Lines 227-228**:

‘DBS-induced entrainment is modulated by movement and is present in patients that had better performance in a motor task’

Discussion

13. Authors state “the peak frequency of spontaneous FTG gradually decreased until it locked to ½ DBS frequency,” though as noted above this claim should be more clearly supported by the data.

Accordingly, we statistically compared the average modulus of the analytic signal of the spontaneous FTG activity 10 seconds during DBS off, and 10 seconds after DBS was switched on with a LME model, while correcting for repeated measures. FTG after DBS was on, was significantly reduced ($p < 0.001$).

Page 6-7: ‘Results’ (**Lines 137-138**):

‘There is a gradual decrease of spontaneous FTG frequency shortly after switching DBS on ($P < 0.001$)’.

14. “This observation fits the hypothesis that entrainment depends on the presence of endogenous oscillatory activity belonging to physiological processes”, how do the authors explain the presence of DBS induced FTG in patients without apparent endogenous off-stim FTG?

This is a valid point raised by the reviewer and we acknowledge that our initial statement is not very accurate. The occurrence of DBS-induced 1:2 entrainment is facilitated by, rather than dependent on, the presence of endogenous FTG, a hypothesis that has been validated before.²⁴ In our interpretation, supratherapeutic levodopa is related to spontaneous FTG (that is often associated with dyskinesia), leading to more consistent entrainment. However, in a subset of patients ($N = 7$) there was no apparent endogenous FTG during DBS-off but we observed gamma entrainment. We offer two possible explanations of this:

- a) Subthreshold FTG may have been present that still facilitates entrainment at gamma frequency. FTG could potentially induced in those patients at a higher dosage of levodopa;
- b) The Percept IPG's sampling rate (250 Hz) and amplification factor, designed primarily for beta activity detection, may have been too low to capture FTG;

We have re-phrased this sentence accordingly (Discussion, Page 10, **Lines 212-213**):

'This observation fits the hypothesis that entrainment is facilitated by the presence of endogenous oscillatory activity belonging to physiological processes'.

We further refer to the introduction of the manuscript, where we describe the different gamma phenomena and the way they appear in intracranial recordings.

15. "Thus, DBS-induced entrainment may enhance (patho)-physiological oscillatory activity, and this may explain part of the prokinetic effect of subthalamic high-frequency DBS and its effectiveness in alleviating bradykinesia." Are the authors referring to low frequency DBS entrainment or now back to HF DBS? If HF DBS, rather than enhance I believe it should say suppress.

This is a misleading statement, as the hypothesis here is that DBS-induced gamma entrainment potentially enhances physiological activity in the broader gamma band (60-90 Hz) which is thought to have a prokinetic effect.

We correct the following in Discussion (page 11, **Lines 258-259**):

'In our study, high frequency DBS-induced entrainment may enhance physiological oscillatory activity within the broadband gamma, and this may explain part of the prokinetic effect of subthalamic high-frequency DBS and its effectiveness in alleviating bradykinesia'.

16. Can authors comment on stimulation delivered in the gamma range (60-80Hz), which a number of studies have explored? Based on this entrainment hypothesis, should we expect that stimulation delivered at gamma band frequencies to produce therapeutic effects by enhancing prokinetic gamma signals? Or is there something special about the 1:2 entrainment that produces greater effects than 1:1 entrainment?

Low-frequency stimulation (LFS; <130 Hz), particularly in the gamma range, has been shown to improve axial symptoms, including freezing of gait²⁸⁻³⁰, as well as speech and balance.³¹ However, the literature in this domain remains heterogeneous, with some studies suggesting that the benefits of LFS on axial symptoms are short-term³² or not finding clear evidence to support the superiority of LFS over high-frequency stimulation (HFS).³³ Despite this, stimulation at 130 Hz remains more effective for treating bradykinesia and rigidity—key levodopa-responsive symptoms that were predominant in the cohort investigated in our study.²⁵

A potentially interesting avenue for future research would be to apply DBS at double the patient's specific spontaneous FTG. The hypothesis is that this approach might induce a 1:2 gamma entrainment with strong prokinetic effects that might lead to dyskinesia. A similar mechanism has been demonstrated for tremor, where a larger coherence peak appeared at the patient's tremor frequency and its first harmonic.³⁴ Here, due to hardware limitations with the PERCEPT PC IPG, which restricts the frequencies that can be used for stimulation while recording LFPs, we were unable to explore this idea and therefore cannot answer this question.

In conclusion, from a clinical perspective, we know that 130 Hz stimulation is more effective for managing PD symptoms, particularly bradykinesia and rigidity. For this reason, the DBS-induced 1:2

gamma entrainment appears to have a unique effect. Devices with greater flexibility in adjusting DBS parameters while recording LFPs could make it possible to explore the potential benefits of different DBS frequencies in more depth. For this reason, we restrain from commenting lower frequency stimulation in our manuscript.

17. “Our data suggest that this biomarker is not solely associated with dyskinesia but could be assigned to a prokinetic state that should be maintained.” There is high inter-subject variability with FTG, and some patient’s optimal clinical setting is lower than when FTG appears, others greater. Can authors comment on the potential patient specific characterization that would likely be required to employ adaptive algorithms that utilize FTG?

This study is the first to investigate such a large patient cohort with the specific aim of characterizing DBS-induced 1:2 entrainment. Although our findings offer valuable insights, there are various patient-specific factors, such as differences in DBS amplitude, dopamine levels, and contact selection, as well as directional stimulation, may influence the signal. Future research should systematically examine these variables to further clarify their effects on the signal. We added the following statement to the Discussion (Page 13, **Lines 319-325**), as per the comment 8 of Reviewer 1:

‘Future studies could aim to chronically record 1:2 gamma entrainment while patients are at home, allowing for the assessment of neural activity during varying levels of dopaminergic medication intake, daily activities, and dyskinesia severity. This approach could help optimize its potential for use amplitude modulation in closed-loop DBS systems. Additionally, the influence of contact selection for stimulation could be investigated to disentangle other DBS parameters that influence 1:2 entrainment’.

18. “We show that the amplitude of the entrainment increased during movement and that motor performance is improved with entrainment.” I think this slightly overgeneralizes the findings. To me it a more accurate way to describe the results is that it appears that the amplitude of the entrainment increased during movement in patients with DBS-induced FTG, and that motor performance was better in that group than patients without DBS-induced FTG. As mentioned earlier, if authors were able find correlations within subject and/or across subjects regarding level of entrainment relative to degree of motor improvement, that would provide stronger evidence to make this claim.

Since we show that there is no linear relationship between the power of entrainment and motor improvement we rephrase as follows (Page 14: ‘Conclusion’, **Lines 356-358**):

‘We show that the amplitude of the entrainment increased during movement and that motor performance was improved in the group with than in the group without entrainment’.

This is also commented in the discussion (Page 11: ‘Discussion’, **Lines 249-254**):

“More importantly, we show that patients that exhibited DBS-induced gamma entrainment performed better in the finger tapping task than the patients that did not. Although entrainment amplitude did not correlate with motor improvement, we hypothesize that entrainment per se could facilitate the physiological mechanism that is related to the 1:2 entrainment frequency, in this case likely the prokinetic gamma synchronization.”

and Page 10: ‘Discussion’ title, **Lines 227-228**:

'DBS-induced entrainment is modulated by movement and is present in patients that had better performance in a motor task'

Reviewer #3 (Remarks to the Author):

This is a very interesting, well-written manuscript investigating half-harmonic entrainment in patients with Parkinson's disease. The study boasts a relatively large sample size for this type of study and, crucially, relates half-harmonic entrainment to motor performance, thereby providing evidence of its clinical relevance.

The manuscript is to the point, and I only have minor comments.

-I would be grateful if the authors could add line numbers in future submissions (and refer to them when listing their changes)

We thank the reviewer for the constructive comments. Below a point-by-point response with added line numbers corresponding to the changes on the manuscript.

-Introduction

1. Ref 8 does not show 1:2 entrainment in the off-medication state. The signals measured do not occur at half the frequency of stimulation, so are not consistent with entrainment. Could the authors please clarify or remove the statement.

We kindly refer to comment #2 from Reviewer 2, where we explain the rationale of adding this citation. We recognize that this is a confusing reference, so we subsequently removed it from Page 3, 'Introduction' (Lines 32-34):

'DBS-induced entrainment was identified in a few patients as a narrow-band activity with a peak at $\frac{1}{2}$ of the DBS frequency during the on-dopaminergic state.'

-Results

2. Fig 1B: in a number of patients the stim amplitude was not increased much. 1:2 entrainment may have disappeared in these patients at higher stimulation amplitudes. I would suggest mentioning this in the limitation section.

The following limitation has added to the manuscript (Page 13-14: 'Limitations', Lines: 343-347):

'Fourth, in many patients the stimulation amplitude was not increased substantially, as it was cut-off to the level in which side-effects were tolerated. This did not allow us to investigate, whether the power of entrainment would decrease with higher stimulation intensities in a larger number of patients (apart from the three cases described here), in line with the assumptions of the Arnold Tongue framework.'

3. How do the authors determine whether entrainment is present or not in the data?

As mentioned in the manuscript, the presence/peaks of gamma activity were selected visually, as they were distinct narrow-band peaks. For a more accurate and automatic detection of entrainment, we retrospectively checked for the presence of peaks using the algorithm 'findpeaks' in Python (threshold = 0.03). Peaks were detected in STNs #1-#15 in exactly half of the DBS frequency. In the figure below we show an example STN for each subcohort, as presented in the manuscript.

This is now added to the Methods section of the manuscript (page 16, Lines 411-412):

'The peak selection was confirmed by using 'findpeaks in Python (threshold = 0.03)'.

Figure 3. Peak detection of subharmonic activity. Example power spectra of each subcohort (left with spontaneous FTG, middle without spontaneous FTG, right without gamma activity) and their corresponding peaks -if detected- denoted with the black marker.

4. Fig 2B: I suppose the authors are using the average modulus of the analytic signal, not the average analytic signal (which is a timeseries of complex numbers). This should be fixed in the figure/caption/main text.

We thank the reviewer for this important mathematical distinction. In the current analysis we use the average modulus of the analytic signal, showing only the amplitude (envelope) of the original signal, and not the complex values. The manuscript was changed accordingly in the following parts:

- Results, page 6, **Line 134:** *'Figure 2b shows three averaged envelopes of the analytic signals over time (n = 8 STNs).'*
- Methods, page 16, **Lines 417-428:** *'Next, we took the envelope analytical signal (absolute values), z-scored it to allow for an inter-subject comparison, and smoothed it with a moving average of 2 seconds. For the analysis, we divided the recordings of subjects with entrainment into three phases: DBS-OFF, DBS-ON before the entrainment onset, and DBS-ON after the entrainment onset. Since these phases have varying durations across patients, we resized each phase to a common duration by resampling the smoothed z-scored envelope of the analytic signal. Resampling the z-scored envelope preserved the overall dynamics of the signal, while allowing for group level analyses'.*
- Caption Figure 2B: *'Averaged envelope of the analytic signal (STNs #1-8) of spontaneous FTG peak as identified in the DBS-off state (blue), of the 1:2 entrainment frequency band as identified during DBS-ON (pink), and of the intermediate activity (green) with increased DBS intensity (grey).'*
- Accordingly, this was corrected on the label of the y axis of figure 2B.

5. Given the clinical data in Table 1, the authors may be able to check if the clinical benefit of DBS was lower in patients who did not display entrainment?

Please refer to Comment 4 from Reviewer #1 for a detailed comparison of UPDRS-III improvements between the subgroup of patients who exhibited entrainment and those who did not. The comparisons cover both total UPDRS-III scores and contralateral hemi-body subscales.

6. Bottom of page 7: LME abbreviation is undefined.

The abbreviation is clarified (Page 8: 'Results', **Line 173**):

'DBS amplitude had no significant effect as a fixed effect on these four metrics in Linear Mixed Effects Models (LMEs; Figure 3c)'.

Discussion

7. Page 8: "that there are more, previously unexplored mechanisms" >> more is strange here, can simply be deleted.

This is accordingly changed in Page 8: 'Discussion', **Lines 195-196:**

'These findings suggest that there are previously unexplored mechanisms of DBS'.

8. Page 9: "This observation fits the hypothesis that entrainment depends on the presence of endogenous oscillatory activity". The hypothesis that entrainment is more likely when endogenous oscillator activity is present was also supported by modelling in a recent letter (<https://doi.org/10.1016/j.brs.2024.02.017>).

We thank the reviewer for the added citation to our Discussion. In this Editor's letter the authors confirm with the use of a computational model (coupled Kuramoto oscillators) that pre-existing oscillatory activity is essential for a subharmonic (1:2) entrainment. We add this citation to our manuscript. We have clarified the statement above, according to Comment #14 of Reviewer 2, to account for the substantial number of STNs (N = 7), which did not have a detectable FTG activity during DBS-off:

Page 9: 'Discussion' (**Lines 212-213**):

This observation fits the hypothesis that entrainment is facilitated by the presence of endogenous oscillatory activity belonging to physiological processes.^{10,24,35,36}

9. Page 9: "Entrainment caused by subthalamic DBS would typically appear in the gamma band around stimulation frequency (130 Hz) or its subharmonics (60-65 Hz)." >> the frequency ranges at 1:1 and 1:2 should be made consistent.

With the PERCEPT, when stimulating and recording simultaneously, the DBS frequency is 130 Hz for 3389 electrodes, but is restricted to 125 Hz for the SenSight. We clarify the sentence (Page 11: 'Discussion', **Lines 243-244**):

'Entrainment caused by subthalamic DBS would typically appear in the gamma band at stimulation frequency (i.e. 130 Hz) or its subharmonics (65 Hz).'

-Methods

10. Contact selection is based on beta in 9 patients, but on gamma entrainment in the rest of the patients. In the former case, are patients less likely to show gamma entrainment?

The reviewers question prompted us to an interesting observation, i.e. the 4 patients without entrainment had i) ring electrodes and ii) contact selection was not systematically tested for entrainment. Further studies need to clarify if beta peaks and gamma entrainment co-localize but also if more focal stimulation at segmented contacts is more likely to induce gamma entrainment. We decided not to discuss this phenomenon as we have only 4 patients without entrainment and did not want to overinterpret our observation.

11. Stimulation is unilateral, how were the hemispheres used in the study selected?

From the cohort included in the current study all SenSight patients (N=10) were recruited (hemisphere & contact) based on highest gamma. All 3389 patients (N=9) were retrospectively added and were part of previously published studies.^{25,26} These were selected based on the hemisphere and contact with the highest beta.

12. “Since these phases have varying durations, we resized each phase to a common duration by resampling the smoothed z-scored analytic signal. Resampling the z-scored analytical signals preserved the overall dynamics of the signal, while allowing for group level analyses.” >> is this because the phases have varying durations across patients (rather within patients)?

The durations of these phases (DBS-Off, DBS-On before entrainment onset, and DBS-On after entrainment onset) were different across patients, as the amplitude of entrainment onset was patient specific. This is clarified within the text (Page 16: ‘Methods’, **Line 425**):

‘Since these phases have varying durations across patients, we resized each phase to a common duration’.

13. Page 15: “We further investigated the effects of varying DBS settings on three different gamma phenomena, i.e., FTG without stimulation...” >> the formulation implies that DBS would have an effect on FTG without stimulation, I would suggest to rephrase this.

This is accordingly rephrased (Page 16: ‘Methods’, **Lines 413-416**):

‘We further investigated the effect of stimulation on three signals filtered around different peaks, i.e., around the patient specific peak of FTG (referred to as ‘spontaneous FTG’), around the 1:2 entrainment peak, and taking the intermediate activity between these two’.

References

1. Swann, N. C., de Hemptinne, C., Miocinovic, S., Qasim, S., Wang, S. S., Ziman, N., Ostrem, J. L., San Luciano, M., Galifianakis, N. B. & Starr, P. A. Gamma Oscillations in the Hyperkinetic State Detected with Chronic Human Brain Recordings in Parkinson's Disease. *J. Neurosci.* **36**, 6445–6458 (2016).
2. Güttler, C., Altschüler, J., Tanev, K., Böckmann, S., Haumesser, J. K., Nikulin, V. V., Kühn, A. A. & van Riesen, C. Levodopa-Induced Dyskinesia Are Mediated by Cortical Gamma Oscillations in Experimental Parkinsonism. *Mov. Disord.* **36**, 927–937 (2021).
3. Jenkinson, N., Kühn, A. A. & Brown, P. Gamma oscillations in the human basal ganglia. *Exp. Neurol.* **245**, 72–76 (2013).
4. Wiest, C., Torrecillos, F., Tinkhauser, G., Pogosyan, A., Morgante, F., Pereira, E. A. & Tan, H. Finely-tuned gamma oscillations: Spectral characteristics and links to dyskinesia. *Exp. Neurol.* **351**, 113999 (2022).
5. Sermon, J. J., Olaru, M., Ansó, J., Cernera, S., Little, S., Shcherbakova, M., Bogacz, R., Starr, P. A., Denison, T. & Duchet, B. Sub-harmonic entrainment of cortical gamma oscillations to deep brain stimulation in Parkinson's disease: Model based predictions and validation in three human subjects. *Brain Stimulat.* **16**, 1412–1424 (2023).
6. Duchet, B., Sermon, J. J., Weerasinghe, G., Denison, T. & Bogacz, R. How to entrain a selected neuronal rhythm but not others: open-loop dithered brain stimulation for selective entrainment. *J. Neural Eng.* **20**, 026003 (2023).
7. Lofredi, R., Neumann, W.-J., Bock, A., Horn, A., Huebl, J., Siegert, S., Schneider, G.-H., Krauss, J. K. & Kühn, A. A. Dopamine-dependent scaling of subthalamic gamma bursts with movement velocity in patients with Parkinson's disease. *eLife* **7**, e31895 (2018).
8. Gilron, R., Little, S., Perrone, R., Wilt, R., de Hemptinne, C., Yaroshinsky, M. S., Racine, C. A., Wang, S. S., Ostrem, J. L., Larson, P. S., Wang, D. D., Galifianakis, N. B., Bledsoe, I. O., San Luciano, M., Dawes, H. E., Worrell, G. A., Kremen, V., Borton, D. A., Denison, T. & Starr, P. A. Long-term wireless streaming of neural recordings for circuit discovery and adaptive stimulation in individuals with Parkinson's disease. *Nat. Biotechnol.* **39**, 1078–1085 (2021).
9. Olaru, M., Cernera, S., Hahn, A., Wozny, T. A., Anso, J., de Hemptinne, C., Little, S., Neumann, W.-J., Abbasi-Asl, R. & Starr, P. A. Motor network gamma oscillations in chronic home recordings predict dyskinesia in Parkinson's disease. *Brain awae004* (2024). doi:10.1093/brain/awae004
10. Coffey, E. B. J., Arseneau-Bruneau, I., Zhang, X., Baillet, S. & Zatorre, R. J. Oscillatory Entrainment of the Frequency-following Response in Auditory Cortical and Subcortical Structures. *J. Neurosci.* **41**, 4073–4087 (2021).
11. Johnson, T. D., Keefe, K. R. & Rangel, L. M. Stimulation-induced entrainment of hippocampal network activity: Identifying optimal input frequencies. *Hippocampus* **33**, 85–95 (2023).
12. Ozen, S., Sirota, A., Belluscio, M. A., Anastassiou, C. A., Stark, E., Koch, C. & Buzsáki, G. Transcranial Electric Stimulation Entrained Cortical Neuronal Populations in Rats. *J. Neurosci.* **30**, 11476–11485 (2010).
13. Ross, B., Tremblay, K. L. & Alain, C. Simultaneous EEG and MEG recordings reveal vocal pitch elicited cortical gamma oscillations in young and older adults. *NeuroImage* **204**, 116253 (2020).
14. Makeig, S., Westerfield, M., Jung, T.-P., Enghoff, S., Townsend, J., Courchesne, E. & Sejnowski, T. J. Dynamic Brain Sources of Visual Evoked Responses. *Science* **295**, 690–694 (2002).
15. Dohrmann, K., Weisz, N., Schlee, W., Hartmann, T. & Elbert, T. in *Prog. Brain Res.* **166**, 473–554 (Elsevier, 2007).
16. Zoefel, B., Archer-Boyd, A. & Davis, M. H. Research data supporting 'Phase entrainment of brain oscillations causally modulates neural responses to intelligible speech'. (2017). doi:10.17863/CAM.16677

17. Dinner, D. S., Neme, S., Nair, D., Montgomery, E. B., Baker, K. B., Rezai, A. & Lüders, H. O. EEG and evoked potential recording from the subthalamic nucleus for deep brain stimulation of intractable epilepsy. *Clin. Neurophysiol.* **113**, 1391–1402 (2002).
18. Walker, H. C., Huang, H., Gonzalez, C. L., Bryant, J. E., Killen, J., Cutter, G. R., Knowlton, R. C., Montgomery, E. B., Guthrie, B. L. & Watts, R. L. Short latency activation of cortex during clinically effective subthalamic deep brain stimulation for Parkinson's disease. *Mov. Disord.* **27**, 864–873 (2012).
19. Awad, M. Z., Vaden, R. J., Irwin, Z. T., Gonzalez, C. L., Black, S., Nakhmani, A., Jaeger, B. C., Bentley, J. N., Guthrie, B. L. & Walker, H. C. Subcortical short-term plasticity elicited by deep brain stimulation. *Ann. Clin. Transl. Neurol.* **8**, 1010–1023 (2021).
20. Johnson, L. A., Wang, J., Nebeck, S. D., Zhang, J., Johnson, M. D. & Vitek, J. L. Direct Activation of Primary Motor Cortex during Subthalamic But Not Pallidal Deep Brain Stimulation. *J. Neurosci.* **40**, 2166–2177 (2020).
21. Steiner, L. A., Crompton, D., Sumarac, S., Vetkas, A., Germann, J., Scherer, M., Justich, M., Boutet, A., Popovic, M. R., Hodaie, M., Kalia, S. K., Fasano, A., Hutchison Wd, W. D., Lozano, A. M., Lankarany, M., Kühn, A. A. & Milosevic, L. Neural signatures of indirect pathway activity during subthalamic stimulation in Parkinson's disease. *Nat. Commun.* **15**, 3130 (2024).
22. Johnson, K. A., Cagle, J. N., Lopes, J. L., Wong, J. K., Okun, M. S., Gunduz, A., Shukla, A. W., Hilliard, J. D., Foote, K. D. & De Hemptinne, C. Globus pallidus internus deep brain stimulation evokes resonant neural activity in Parkinson's disease. *Brain Commun.* **5**, fcd025 (2023).
23. Oehr, C. R., Cernera, S., Hammer, L. H., Shcherbakova, M., Yao, J., Hahn, A., Wang, S., Ostrem, J. L., Little, S. & Starr, P. A. Chronic adaptive deep brain stimulation versus conventional stimulation in Parkinson's disease: a blinded randomized feasibility trial. *Nat. Med.* (2024). doi:10.1038/s41591-024-03196-z
24. Sermon, J. J., Starr, P. A., Denison, T. & Duchet, B. Pre-existing oscillatory activity as a condition for sub-harmonic entrainment of finely tuned gamma in Parkinson's disease. *Brain Stimulat.* **17**, 488–490 (2024).
25. Feldmann, L. K., Lofredi, R., Neumann, W.-J., Al-Fatly, B., Roediger, J., Bahnert, B. H., Nikolov, P., Denison, T., Saryyeva, A., Krauss, J. K., Faust, K., Florin, E., Schnitzler, A., Schneider, G.-H. & Kühn, A. A. Toward therapeutic electrophysiology: beta-band suppression as a biomarker in chronic local field potential recordings. *Npj Park. Dis.* **8**, (2022).
26. Mathiopoulou, V., Lofredi, R., Feldmann, L. K., Habets, J., Darcy, N., Neumann, W.-J., Faust, K., Schneider, G.-H. & Kühn, A. A. Modulation of subthalamic beta oscillations by movement, dopamine, and deep brain stimulation in Parkinson's disease. *Npj Park. Dis.* **10**, 77 (2024).
27. Adam, E. M., Brown, E. N., Kopell, N. & McCarthy, M. M. Deep brain stimulation in the subthalamic nucleus for Parkinson's disease can restore dynamics of striatal networks. *Proc. Natl. Acad. Sci.* **119**, e2120808119 (2022).
28. Conway, Z. J., Silburn, P. A., Perera, T., O'Malley, K. & Cole, M. H. Low-frequency STN-DBS provides acute gait improvements in Parkinson's disease: a double-blinded randomised cross-over feasibility trial. *J. NeuroEngineering Rehabil.* **18**, 125 (2021).
29. Di Giulio, I., Kalliolia, E., Georgiev, D., Peters, A. L., Voyce, D. C., Akram, H., Foltynie, T., Limousin, P. & Day, B. L. Chronic Subthalamic Nucleus Stimulation in Parkinson's Disease: Optimal Frequency for Gait Depends on Stimulation Site and Axial Symptoms. *Front. Neurol.* **10**, 29 (2019).
30. Ramdhani, R. A., Patel, A., Swope, D. & Kopell, B. H. Early Use of 60 Hz Frequency Subthalamic Stimulation in Parkinson's Disease: A Case Series and Review. *Neuromodulation Technol. Neural Interface* **18**, 664–669 (2015).
31. Zibetti, M., Moro, E., Krishna, V., Sammartino, F., Picillo, M., Munhoz, R. P., Lozano, A. M. & Fasano, A. Low-frequency Subthalamic Stimulation in Parkinson's Disease: Long-term Outcome and Predictors. *Brain Stimulat.* **9**, 774–779 (2016).
32. Vijjaratnam, N., Girges, C., Wirth, T., Grover, T., Preda, F., Tripoliti, E., Foley, J., Scelzo, E., Macerollo, A., Akram, H., Hyam, J., Zrinzo, L., Limousin, P. & Foltynie, T. Long-term success of low-

frequency subthalamic nucleus stimulation for Parkinson's disease depends on tremor severity and symptom duration. *Brain Commun.* **3**, fcab165 (2021).

33. Vallabhajosula, S., Haq, I. U., Hwynn, N., Oyama, G., Okun, M., Tillman, M. D. & Hass, C. J. Low-frequency Versus High-frequency Subthalamic Nucleus Deep Brain Stimulation on Postural Control and Gait in Parkinson's Disease: A Quantitative Study. *Brain Stimulat.* **8**, 64–75 (2015).

34. Hirschmann, J., Hartmann, C. J., Butz, M., Hoogenboom, N., Özkurt, T. E., Elben, S., Vesper, J., Wojtecki, L. & Schnitzler, A. A direct relationship between oscillatory subthalamic nucleus–cortex coupling and rest tremor in Parkinson's disease. *Brain* **136**, 3659–3670 (2013).

35. Hemptinne, C., Wang, D. D., Miocinovic, S., Chen, W., Ostrem, J. L. & Starr, P. A. Pallidal thermolesion unleashes gamma oscillations in the motor cortex in Parkinson's disease. *Mov. Disord.* **34**, 903–911 (2019).

36. Notbohm, A., Kurths, J. & Herrmann, C. S. Modification of Brain Oscillations via Rhythmic Light Stimulation Provides Evidence for Entrainment but Not for Superposition of Event-Related Responses. *Front. Hum. Neurosci.* **10**, (2016).

Point-by-point response to Reviewers

Reviewer #1 (Remarks to the Author):

The authors did a great job responding to my comments. Congratulations on this important work.

We thank the reviewer for the constructive comments and questions.

Reviewer #2 (Remarks to the Author):

Overall the authors provided very thorough and adequate response to reviewer comments. Changes and additions made have strengthened the manuscript.

One remaining point:

Regarding response to Reviewer 2 Comment 16, and discussion paragraph beginning with line 249, I do still think it would be useful to include some discussion about potential gamma entrainment with gamma range stimulation. At the end of their response to Comment 16, they state "we restrain from commenting [on] lower frequency stimulation in our manuscript." The authors state, however, in the discussion Line 254 that "...This hypothesis is supported by previous studies that showed that stimulation at 20 Hz (i.e. within the beta band of 13-35 Hz) worsened motor performance and bradykinesia. In this case, entrainment possibly promoted activity within the beta frequency range that is considered pathological and enhanced parkinsonian symptomatology."

Given what the authors described as likely entrainment in the beta band range during beta band (20hz) stimulation (I presume they are considering likely producing 1:1 entrainment), which they state supports their hypothesis that "entrainment per se could facilitate the physiological mechanism that is related to the 1:2 entrainment frequency, in this case likely the prokinetic gamma synchronization," would it not also make sense to assume that gamma band (1:1) entrainment is likely occurring during stimulation delivered at gamma band frequencies as performed in previous studies? Are those studies not relevant to the entrainment hypothesis, similar to the 20hz stimulation studies? It seems to me an interesting unanswered question to raise in this discussion section why motor signs like bradykinesia would not be particularly improved during gamma range stimulation. Based on the entrainment hypothesis that gamma synchronization is prokinetic, readers might think it would make sense to stimulate and entrain at gamma frequencies, though as the authors noted in their response from a clinical perspective this is not very effective. It is my opinion that some additional discussion of this topic would be warranted in this section.

We understand the interest in 1:1 gamma entrainment of the referee. Nevertheless, our study did not test individual gamma frequency nor stimulation at 70 or 80 Hz to provide evidence for 1:1 entrainment. We have now added the following paragraph to clarify that there is no clinical evidence yet for gamma band stimulation to specifically improve bradykinesia. Indeed, there might be a different mechanism underlying the 1:2 entrainment that is associated with better motor performance. Future studies are needed to specifically explore 1:1 and 1:2 gamma entrainment in PD patients.

Discussion, Page 11:

'Interestingly, stimulation at individual gamma frequency (70-90 Hz) that could be assumed to lead to 1:1 entrainment has not been described to specifically improve bradykinesia but showed similar results to standard high frequency stimulation.¹ Clinically, 70-90 Hz stimulation is often used to improve dyskinesia, dysarthria or freezing in some patients.²⁻⁴ Further research is needed to clarify the underlying mechanism for 1:2 gamma entrainment, potentially representing an additional mechanism of DBS. Our data suggest a potential utility of 1:2 entrainment as an informative signal when adjusting DBS amplitudes for patients with PD.'

Reviewer #3 (Remarks to the Author):

I would like to thank the authors for having appropriately addressed my comments. Two minor points:

1) The more common terminology is the analytic signal, not the analytical signal.

This was corrected in the case when it was misstated in 'Methods' (page 16):

'Next, we took the envelope analytic signal (absolute values)'

2) It would be good to caveat the statement added Lines 103-108 by mentioning that the test was underpowered (n=4 in group without entrainment).

The following statement has been added:

'Nevertheless, the test is limited by the small sample size of the group without entrainment (N=4).'

References

1. Tsang, E. W. *et al.* Subthalamic deep brain stimulation at individualized frequencies for Parkinson disease. *Neurology* **78**, 1930–1938 (2012).
2. Conway, Z. J., Silburn, P. A., Perera, T., O'Maley, K. & Cole, M. H. Low-frequency STN-DBS provides acute gait improvements in Parkinson's disease: a double-blinded randomised cross-over feasibility trial. *J. NeuroEngineering Rehabil.* **18**, 125 (2021).
3. Di Giulio, I. *et al.* Chronic Subthalamic Nucleus Stimulation in Parkinson's Disease: Optimal Frequency for Gait Depends on Stimulation Site and Axial Symptoms. *Front. Neurol.* **10**, 29 (2019).
4. Ramdhani, R. A., Patel, A., Swope, D. & Kopell, B. H. Early Use of 60 Hz Frequency Subthalamic Stimulation in Parkinson's Disease: A Case Series and Review. *Neuromodulation Technol. Neural Interface* **18**, 664–669 (2015).